# Various lithospheric deformation patterns derived from rheological contrasts between continental terranes: Insights from 2-D numerical simulations

Renxian Xie[1, 2], Lin Chen[3], Jason P. Morgan[2], Yongshun John Chen[2*]

[1]School of Transportation Engineering, East China Jiaotong University, Nanchang, 330013, China

[2]Department of Ocean Science and Engineering, Southern University of Science and Technology, Shenzhen, 518055, China

[3]State Key Laboratory of Lithospheric Evolution, Institute of Geology and Geophysics, Chinese Academy of Sciences, Beijing, 100029, China

*Correspondence to*: Yongshun John Chen (johnyc@sustech.edu.cn)

**Abstract.** Continents are formed by the amalgamation of numerous micro-terranes and island arcs, so they have spatially varying lithosphere strengths. The Crème brûlée (CB) model and the Jelly sandwich (JS) model have been commonly used to describe continental lithosphere strength-depth variations. Depending on the strength of continental lower crust, the CB and JS models can be further subdivided into two subclasses, in which the I subclass (CB-I and JS-I) and II subclass (CB-II and JS-II) respectively have a strong or weak lower crust. During continental collision, lithosphere deformation is the byproduct of the comprehensive interaction of multiple terranes. Here we used 2-D thermo-mechanical numerical models that contain three continental terranes to systematically explore the effects of terranes with various strengths on continental deformation, and studied the effects of different rheological assumptions on terrane deformation. We found four types of lithosphere deformation patterns: collision, subduction, thickening and delamination, and replacement. These simulation patterns are seen in observed deformation patterns and structures in East Asia, suggesting they are likely to be naturally occurring modes of intracontinental orogenesis.

## 1. Introduction

Continents have undergone multiple break-up and assembly events during the past ~2 billion years, with the assembly events often being associated with the accretion and deformation of numerous micro-terranes (Mitchell et al., 2021). Accreted terranes have different ages ranging from ~3500 – 3000 Ma to 50 – 0 Ma, and diverse compositions and structures linked to their diverse continental, arc, or

oceanic origin, which often leads to them having distinct initial lithospheric thicknesses and strengths
(Artemieva, 2006; Audet and Bürgmann, 2011; Pasyanos et al., 2014; Morgan and Vannucchi, 2022).
The lithosphere of ancient continental terranes like cratons are usually thick and strong, while younger
lithosphere of continental margins and tectonically active regions is thin and weak (Audet and
Bürgmann, 2011; Burov, 2011), and deeply buried former oceanic fragments can have temperature and
strengths that vary over ~0.5 Gyr timescale (Morgan and Vannucchi, 2022).
Continental lithosphere strength conventionally been represented by two prevailing rheology models
—the Crème brûlée (CB) and the Jelly sandwich (JS) idealizations (Chen and Molnar, 1983; Jackson,
2002; Burov and Watts, 2006; Bürgmann and Dresen, 2008; Burov, 2011). The Crème brûlée scenario
suggests that lithosphere strength resides entirely in the crust, with the lithospheric mantle being much
weaker (with this strength contrast being the explanation for why little seismicity is typically seen in
the continental mantle, despite rock-mechanics arguments that it should usually be stronger than its
overlying crust). In contrast, the Jelly sandwich model is based on conventional rock mechanics
arguments which imply that in general the continental middle and lower crust should be weaker than
overlying cooler upper crust and underlying further-from-solidus lithospheric mantle (Figure 1a). The
rheology of the continental lower crust can also differ strongly in different continental terranes due to
the varieties in composition, temperature, water content, stress, and tectonic environment (Bürgmann
and Dresen, 2008; Hacker et al., 2015; Morgan and Vannucchi, 2022). Therefore, the CB and JS
conceptualizations can be further subdivided into CB-I and CB-II, JS-I, and JS-II subclasses that reflect
potentially variable strengths of the lower crust: CB-I and JS-I, CB-II and JS-II have strong and weak
continental lower crust, respectively (Fig. 1a). Observations in Eastern Asia show a wide variability in
terrane deformation styles that argue for the potential feasibility of all four of these rheological models
(Figure 1b).
Several previous numerical modelling studies have discussed the effects of rheological contrasts
between terranes in lithosphere deformation in a collisional system. Studies containing two terranes
have explored contrasts in crustal rheology, and found that this can greatly change the morphology, size
and deep lithosphere structure of collisional orogenic belt (Chen, 2021; Chen et al., 2017; Cook and
Royden, 2008; Faccenda et al., 2008; Sun and Liu, 2018; Vogt et al., 2018; Xie et al., 2021). Strong
crust also has the potential to protect its underlying lithospheric mantle from deformation and
destruction (Heron and Pysklywec, 2016). Studies containing three or more terranes in their models
have usually focused on the middle terrane which can play a crucial role in lithosphere deformation in a
collisional system (Kelly et al., 2016, 2020; Li et al., 2016; Huangfu et al., 2018, 2022; Sun and Liu,
2018; Xie et al., 2023). A weak middle terrane is easy to be thickened, to the point where eventually its
lithospheric mantle can be delaminated from the crust; while a moderate-strength middle terrane can
induce far-field orogenesis; and a strong middle terrane may prevent propagation of deformation and
facilitate underthrusting of the advancing terrane. In addition, some studies have also stressed the
importance of local pre-existing weak zones which can change the order and style of lithosphere
deformation (Chen et al., 2020; Heron et al., 2016; Sokoutis and Willingshofer, 2011; Xie et al., 2021).
Large-scale continental collisional system often involves the multiple units of an indenting terrane, a
middle terrane, and far-end backwall terranes. These terranes have different lithosphere rheologies and
thicknesses, and they collectively contribute to several styles of continental deformation (Artemieva,
2006; Audet and Bürgmann, 2011; Pasyanos et al., 2014; Morgan and Vannucchi, 2022). Here, we use a
2-D thermo-mechanical numerical modeling method to systematically study the effects of terranes with
various rheological properties on continental deformation. Our numerical models simulate a
continent-continent collisional system that contains three continental terranes. They explores the effects
of four groups of lithosphere deformation patterns linked to the four rheological idealizations of CB-I,
CB-II, JS-I, and JS-II applied to each terrane. We will summarize the rheological features for each
deformation pattern, and then apply the simulations to better understand ongoing and past deformation
histories of various orogenic belts in eastern Asia, such as the eastern Tien Shan orogenic belt, the
Tibetan Plateau and the Early Paleozoic Orogen in Southeastern China.
**2. Numerical modelling method and model setup**
**2.1. Numerical modelling method**
Our thermo-mechanical models were performed with the I2VIS code of Gerya and Yuen (2003a),
previously used in Xie et al. (2021, 2023). This code combines finite differences with marker-in-cell
techniques to solve the mass, momentum, and energy conservation equations for incompressible flow.
It incorporates the non-Newtonian visco-plastic rheologies for the lithosphere, as well as the possibility
to include parameterizations of the effects of surface processes like sedimentation and erosion.
**2.1.1. Governing equations**
The mass conservation equation for incompressible flow is:

$$\frac{\partial v_x}{\partial x} + \frac{\partial v_y}{\partial y} = 0 \,, \tag{1}$$

The momentum conservation equations (Stokes equations) are:

$$\begin{aligned}
\frac{\partial \sigma'_{xx}}{\partial x} + \frac{\partial \sigma'_{xy}}{\partial y} &= \frac{\partial P}{\partial x} \\
\frac{\partial \sigma'_{yy}}{\partial y} + \frac{\partial \sigma'_{xy}}{\partial x} &= \frac{\partial P}{\partial y} - g\rho
\end{aligned}\,, \tag{2}$$

The energy (heat) conservation equation is:

$$\begin{aligned}
\rho C_p \frac{DT}{Dt} &= -\frac{\partial q_x}{\partial x} - \frac{\partial q_y}{\partial y} + H_r + H_s + H_a + H_L \\
q_x &= -k\frac{\partial T}{\partial x} \\
q_y &= -k\frac{\partial T}{\partial y} \\
H_a &= T\alpha\frac{DP}{Dt} \\
H_s &= \sigma'_{xx}\dot{\varepsilon}_{xx} + \sigma'_{yy}\dot{\varepsilon}_{yy} + 2\sigma'_{xy}\dot{\varepsilon}_{xy}
\end{aligned}\,, \tag{3}$$

where $x$ and $y$ represent the horizontal and vertical coordinate directions, and $v_x$ and $v_y$ are the
corresponding velocity components, respectively. $\sigma'_{ij}$ and $\dot{\varepsilon}_{ij}$ $(i, j = x, y)$ are deviatoric stress and
strain-rate tensors, respectively; $g$ is the gravitational acceleration; $\rho$ is density. In the heat conservation
equation, $q_x$ and $q_y$ are the horizontal and vertical components of the heat flux, respectively; $C_p$ is heat
capacity, and $H_r$, $H_a$, $H_s$, and $H_L$ denote the radioactive, adiabatic, shear, and latent heat production,
respectively; $k$ is the thermal conductivity.
The rheological constitutive relationship connects the deviatoric stress and strain rate:

$$\begin{aligned}
\sigma'_{xx} &= 2\eta_{eff}\dot{\varepsilon}_{xx}, & \dot{\varepsilon}_{xx} &= \frac{\partial v_x}{\partial x} \\
\sigma'_{xy} &= 2\eta_{eff}\dot{\varepsilon}_{xy}, & \dot{\varepsilon}_{xy} &= \frac{1}{2}\left(\frac{\partial v_x}{\partial y} + \frac{\partial v_y}{\partial x}\right), \\
\sigma'_{yy} &= 2\eta_{eff}\dot{\varepsilon}_{yy}, & \dot{\varepsilon}_{yy} &= \frac{\partial v_y}{\partial y}
\end{aligned} \tag{4}$$

where $\eta_{eff}$ is the effective viscosity.
**2.1.2. Rheology**
Here we make the conventional assumption that the crust and mantle have a visco-plastic rheology.
Viscous deformation is determined as a combination of diffusion and dislocation creep that depends on
temperature, pressure, and strain rate, expressed as (Gerya, 2019):

$$
\begin{aligned}
\eta_{disl} &= \frac{1}{2} \frac{1}{(A_D)^{-1/n}(\dot{\varepsilon}_{II})^{(n-1)/n}} \exp\left(\frac{E_a + V_a P}{nRT}\right) * S \\
\eta_{diff} &= \frac{1}{2} \frac{A_D}{\sigma_{cr}^{(n-1)}} \exp\left(\frac{E_a + V_a P}{RT}\right) * S
\end{aligned}
\tag{5}
$$


For mineral aggregates, both dislocation and diffusion creep occur simultaneously, with a combined
effective viscosity given by:

$$
\frac{1}{\eta_{ductile}} = \frac{1}{\eta_{disl}} + \frac{1}{\eta_{diff}} ,
\tag{6}
$$


where $\eta_{disl}$ and $\eta_{diff}$ are viscosities for dislocation and diffusion creep, respectively. $\sigma_{cr}$ is the
critical stress for the dislocation to diffusion stress transition, and the parameters $A_D$, $E_a$, $V_a$, and $n$ are a
material constant, activation energy, activation volume, and stress exponent, respectively, and $R$ is the
universal gas constant. The strength scaling factor, $S$, is introduced as a simple parameter to vary the
lithospheric viscosity.
Plasticity is implemented using a conventional pseudo-viscous yield criterion first used to study rifting
(e.g. Chen and Morgan, 1990) that is extended to include a strain-weakening-like parameterization of
fracture-related strain weakening (Gerya et al., 2010; Vogt et al., 2017):
$$\eta_{plastic} = \frac{\sigma_{yield}}{2\dot{\varepsilon}_{II}}$$

$$\sigma_{yield} = C + P\phi$$

$$C = \begin{cases} C_a + \left(C_b - C_a\right) \times \dfrac{\gamma}{\gamma_{cr}}, & \text{if } \gamma \leq \gamma_{cr} \\ C_b, & \text{if } \gamma \geq \gamma_{cr} \end{cases} \qquad (7)$$

$$\phi = \begin{cases} \phi_a + \left(\phi_b - \phi_a\right) \times \dfrac{\gamma}{\gamma_{cr}}, & \text{if } \gamma \leq \gamma_{cr} \\ \phi_b, & \text{if } \gamma \geq \gamma_{cr} \end{cases}$$

where $\sigma_{yield}$ is yield stress, $P$ is dynamic pressure, $\gamma$ is the integrated plastic strain, and $\gamma_{cr}$ is the
upper strain limit for fracture-related weakening. $C$ and $\phi$ are cohesion and friction angle that depend
on the plastic value. $C_a$ and $\phi_a$ are the initial and $C_b$ and $\phi_b$ are final strength values, respectively.
This involves making the rheological assumption that deeply percolating fluids and high pore fluid
pressures can significantly lower the plastic strength of fractured rocks.
The final effective viscosity is determined by the minimum value between the ductile and plastic
viscosities (Ranalli, 1995):
$$\eta_{eff} = \min\left(\eta_{ductile}, \eta_{plastic}\right). \qquad (8)$$
**2.1.3. Surface processes**
Topography in our models evolves according to a transport equation that is solved at each time step,
with a crude local parameterization of effects of accounts for sedimentation and erosion:
$$\frac{\partial y_{es}}{\partial t} = v_y - v_x \frac{\partial y_{es}}{\partial x} - v_s + v_e. \qquad (9)$$
Where $y_{es}$ is the vertical position of the surface as a function of horizontal distance $x$; and $v_x$ and $v_y$ are
the corresponding velocity components, respectively. $v_s$ and $v_e$ are the sedimentation and erosion rates,
respectively, conforming to the relation:
$v_s = 0$ mm/yr, $v_e = 0.3$ mm/yr when $y_{es} > 5$ km;
$v_s = 0.3$ mm/yr, $v_e = 0$ mm/yr when $y_{es} < 5$ km.
Owing to surface processes are not our focuses in this study, for the aim of simplification, we adopt a
small erosion and sedimentation rates of 0.3 mm/yr, which are similar to previous studies (Gerya and
Yuan, 2003b; Bian et al., 2020). As well, we simply choose a very large value of 5 km as the threshold
for initiating denudation and sedimentation to further weaken the influences of surface processes on the
evolutions of our model.
**2.2. Model Setup**
The 2-D numerical model covers a rectangular computational domain of 3000 km $\times$ 700 km and
consists of 1360 $\times$ 400 non-uniform grid cells with dozens of mobile markers in each grid cell to
transport physical properties (Figure 2a). Above 300 km, the cell-size of the grid in the middle of
model (X = 1300 – 2200 km) is 1 km $\times$ 1 km, and gradually widens towards the two sides to finally
become 5 km $\times$ 1 km. From the 300 km depth to the model bottom, each grid is stretched to 5 km in the
vertical direction. As a result, the grid in the middle of the model (X = 1300 – 2200 km) is 1 km $\times$ 5 km
and is 5 km $\times$ 5 km in the other regions. Changing resolutions in different model regions can ensure the
model can finely depict lithosphere deformation in the region of interest while improving the
calculation's efficiency.
In the initial configuration, the model comprises three continental terranes — the Pro-, Mid- and
Retro-terrane — which refer to the indenting 'Pro-' terrane driven by plate convergence, an
intermediate 'Mid-' terrane, and a far-end backwall 'Retro-' terrane, respectively (Figure 2a). For the
purpose of simplification, the three terranes are assumed to have the same initial crustal structure with
20 km thick upper and lower crust, respectively. In the meanwhile, to simulate lateral structure
differences within continental lithosphere (Pasyanos et al., 2014), thicknesses of the initial lithospheric
mantle of the Pro-, Mid- and Retro-terrane are 160 km, 90 km, and 120 km, respectively. The rest of
the region is filled by asthenosphere except along the model top, where a 20 km thick layer of "sticky
air" with low viscosity ($1 \times 10^{18}$ Pas) and low density (1 kg/m$^3$) is placed to simulate the effects of a
free surface (Schmeling et al., 2008). Flow laws and material properties for each lithospheric layer are
listed in Table 1.
Mechanical boundary conditions of the model are that the top and sides are free-slip boundaries which
mean that the vertical velocity at the top boundary and horizontal velocity at the side boundaries are all
zero. The bottom is assumed to be a somewhat non-physical 'permeable boundary' that was developed
to reduce the required depth of the computational region (Burg and Gerya, 2005). For top-driven flows
like those considered here, this approximation has been shown to not affect deformation in the upper
parts of the region (Burg and Gerya, 2005). Finally, a constant convergence rate of 20 mm/yr is
assigned to the Pro-terrane (X = 1000 km) to drive the model.
Initial temperature conditions are set as follows: the model top is set to $0\,°C$, the two side boundaries
are adiabatic boundaries with zero horizontal heat fluxes, and the model bottom has an initial
temperature of $1593\,°C$, and can dynamically adjust as the model evolves. The initial thermal gradient
in the crust is $15\,°C/km$ in the three terranes, so their Moho temperature is $600\,°C$. A temperature of
$1330\,°C$ is applied at the bottom of the lithospheric mantle of the three terranes, which leads to the Pro-
and Mid-terrane having minimum and maximal thermal gradients in the lithospheric mantle,
respectively (see the right plane in Figure 2a). An adiabatic thermal gradient of $0.5\,°C/km$ is assumed
within the asthenosphere. The temperature field would evolve over time, thus, although the three
terranes are not in thermal equilibrium at the start of the experiments, it has few effects on model
evolution. The initial setup of lithosphere structure and temperature field make the Mid-terrane weakest
when same rheology model is used for the three terranes.

## 3. Simulation Results

The rheological models of CB-I, CB-II, JS-I, and JS-II result from different strength scaling factors for
the upper crust, the lower crust, and the lithospheric mantle in our numerical models (Figure 2b). We
systematically test the effects of these rheological assumptions on the deformation of the Pro-, Mid-
and Retro-terranes. According to different behaviors of lithosphere deformation, these simulation
results can be categorized into four basic modes of collision, subduction, thickening and delamination,
and replacement (Figures S1 and S2). In the deformation mode of collision, the lithospheric mantle of
the Mid-terrane is extruded out and the lithospheric mantles of the two bounding terranes meet and
collide together. In the deformation mode of subduction, the lithospheric mantle of one of the bounding
terranes subducts into the deep mantle below the Mid-terrane while the other one keeps almost
undeformed. In the deformation mode of thickening and delamination, one of the bounding terranes is
shortened by compression, and delamination may come on the heels of thickening of lithosphere due to
gravitational instability in some cases. In the deformation mode of replacement, the bottom of weak
and thick lithospheric mantle of the bounding terrane is scraped off by the strong lithospheric mantle of
the Mid-terrane, and replaced by the latter. Here, we select a typical case for each mode of lithosphere
deformation to describe more details of these modes of model evolution.

### 3.1. Case 1: Lithosphere Collision

Case 1 represents the scenario of lithosphere collision (Figure 3). In this model, the assumed
rheological models for the Pro-, Mid- and Retro-terrane are JS-I, JS-II, and JS-I, respectively, which
means that the Mid-terrane has a significantly weaker lower crust relative to the Pro- and
Retro-terranes. The lithospheric mantle of the Mid-terrane is also slightly weaker due to its thinner
lithosphere and correspondingly higher initial temperature field. The lithosphere strength profiles of the
three terranes are shown in Figure 3g.
The Mid-terrane is the first to deform when the Pro-terrane begins to collide, absorbing plate
convergence in the form of lithosphere thickening (Figures 3a and 3d). The upper crust of the
Mid-terrane breaks due to strain weakening, and several reverse faults are formed to absorb crustal
shortening. The lower crust folds, and strain diffusely distributes within it. Since the Retro-terrane is
relatively strong, it prevents crustal deformation from propagating into this terrane, and restricts the
bulk of deformation to the Mid-terrane. With continuous advance of the Pro-terrane and resistance of
the Retro-terrane, the crust of the Mid-terrane is intensively shortened, leading to more thrusting
structures in the upper crust and a "flower-like" structure in the lower crust (Figures 3b and 3e). Thrust
structures and crustal deformation also expand toward the Pro- and Retro-terranes at this stage.
Topography also grows towards the two bounding terranes (Figure 7a). Ultimately, the weak
lithospheric mantle of the Mid-terrane is squeezed out, and the Pro- and Retro-terrane's lithospheric
mantles meet and so start to collide beneath the overlying crust of the Mid-terrane (Figures 3c and 3f).

### 3.2. Case 2: Lithosphere Subduction

Case 2 shows lithosphere subduction of the Pro-terrane (Figure 4). In this model, the assumed
rheological models for the Pro-, Mid- and Retro-terrane are JS-II, JS-I, and JS-I, respectively. The
Mid-terrane has a stronger lower crust compared with the Pro-terrane, but its lithospheric mantle is a
little weaker than the Pro-terrane due to higher temperature field resulting from its thinner lithosphere
structure (Figure 4g). When convergence begins, the weak lower crust of the Pro-terrane is blocked by
the stronger lower crust of the Mid-terrane. This induces it to stack in a collisional front to form a
remarkable folding structure (Figures 4a and 4d). The strong lithospheric mantle of the Pro-terrane
continues to move forward and underthrusts beneath the Mid-terrane. As the Pro-terrane advances, its
crust gradually enters the Mid-terrane, inducing shortening and thickening of the upper crust of the
Mid-terrane, while the strong lower crust of the Mid-terrane almost keeps undeformed (Figures 4b and
4e).
Meanwhile, the lithospheric mantle of the Pro-terrane continues to underthrust scraping off part of the
lithospheric mantle of the Mid-terrane. Eventually, the crust of the Pro-terrane wedges a long distance
into the Mid-terrane, and the lithospheric mantle of the Pro-terrane subducts into the deeper mantle
(Figures 4d and 4f). In this example, crustal deformation and topography gradually propagate from the
Pro-terrane to the Mid-terrane, whereas the Retro-terrane remains nominally 'undeformed' at all times
(Figure 7b). In some experiments, the lithospheric mantle of the Retro-terrane can subduct beneath the
Mid-terrane (Figure S1).
**3.3. Case 3: Lithosphere Thickening and Delamination**
Case 3 illustrates the thickening and delamination of the lithospheric mantle of the Pro-terrane (Figure
5). In this case, the rheological models for the Pro-, Mid- and Retro-terranes are CB-II, JS-I, and JS-I,
respectively (Figure 5j). The Pro-terrane has a rheologically weaker lower crust and lithospheric mantle,
making it relatively easy to deform once the collision has started. The lithosphere of the Pro-terrane is
first thickened, and the crust starts to fold in two discrete zones (Figures 5a and 5f). The lower part of
the thickened lithospheric mantle is denser than its ambient mantle owing to lower temperature, which
causes it to drip downwards (Figures 5b–5h). After delamination of the thickened lithosphere,
subduction of the Pro-terrane's lithospheric mantle along one of the deformation localization zones
absorbs the plate convergence (Figures 5e and 5i). Crustal deformation is restricted in the Pro-terrane
until lithosphere delamination, after which crustal strain and topography rapidly spread from the
Pro-terrane to the Mid-terrane (Figure 7c). Like case 2, the Retro-terrane stays essentially undeformed
at all times.
**3.4. Case 4: Lithosphere Replacement**
Case 4 illustrates how the lithospheric mantle of the Pro-terrane is replaced by that of a neighboring
stronger Mid-terrane (Figure 6). In this case, the rheological models for the Pro-, Mid- and
Retro-terranes are CB-I, JS-II, and JS-I, respectively. The Pro-terrane has a strong lower crust and a
thick and weak lithospheric mantle, while the Mid-terrane has a weaker lower crust and a strong
lithospheric mantle (Figure 6g). This lithosphere configuration between the Pro- and the Mid-terrane
causes deformation to be primarily distributed in the Pro-terrane's lithospheric mantle and the
Mid-terrane's crust. As a result, the Mid-terrane's crust becomes intensely shortened by fold and thrust
structures, but its strong lithospheric mantle wedges into the Pro-terrane's thick and weak lithospheric
mantle (Figures 6a, 6b, 6d and 6e). The strong lithospheric mantle of the Mid-terrane scrapes off the
lower part of the weak lithospheric mantle of the Pro-terrane and so replaces it (Figures 6c and 6f).
Similar to case 1, crustal deformation and topography expand from the Mid-terrane towards its side
terranes (Figure 7d). The lithospheric mantle of the Retro-terrane can also be replaced in some cases
(e.g. Figure S1).
**4. Discussion**
**4.1. Rheological Characteristics for Distinct Lithosphere Deformation Patterns**
Distinct lithosphere deformation patterns in our simulations arise from rheological contrasts between
neighboring continental terranes. Figure 8 summarizes the rheological characteristics of these distinct
deformation patterns. When the Mid-terrane's lithospheric mantle is weakest (typified by models in
which the rheological model of the Mid-terrane is CB-II), it is easy for its mantle to be extruded out,
leading to collision between the lithospheric mantles of its surrounding Pro- and Retro-terranes. When
one of the two bounding terranes has extremely weak lithospheric mantle, its lithosphere is first to be
thickened by compression, and delamination may follow due to density-driven instability. When the
lower crust of the Mid-terrane is relatively strong (CB-I or JS-I), while the lower crust is weaker in the
Pro- or Retro-terrane (CB-II or JS-II), the lithospheric mantle of the Pro- or Retro-terrane will tend to
subduct into the deep mantle, e.g. leading to intracontinental subduction. Finally, when the Mid-terrane
has a weak lower crust and strong lithospheric mantle (JS-II), while the Pro- or Retro-terrane has a
strong lower crust and weak lithospheric mantle (CB-I), the lithospheric mantle of the former may
replace the lithospheric mantle of the latter.
When the deformation patterns involve the collision and replacement of lithosphere, continental
deformation involves all three terranes (Figures 3 and 6). In contrast, the other deformation patterns
only involve two terranes, the Pro- or Retro-terrane and the Mid-terrane (Figures 4 and 5). The
rheological properties of the Mid-terrane are responsible for these differences. Like previous numerical
studies (Kelly et al., 2016, 2020; Li et al., 2016; Huangfu et al., 2018, 2022; Sun and Liu, 2018), our
simulations show that a weak Mid-terrane is easier to deform, and that in this case lithosphere
deformation will expand from center to its neighboring sides; while a relatively strong Mid-terrane
prevents deformation from propagating far, so that lithosphere deformation is constrained to occur
within two terranes..
Although our multi-terrane numerical models mainly focus on the impact of the lateral strength
differences between different terranes in a continental collisional system, rheological models of CB-I,
CB-II, JS-I, JS-II also involve vertical rheological variation (Figure 1a). It seems difficult to summarize
how vertical strength variation affects lithosphere deformation of the continental collisional system.
For example, in some cases, only changing the rheological models of the Pro- or Retro-terranes may
produce distinct deformation modes such as collision, subduction, thickening and delamination, and
displacement (e.g., the first and third rows, the third column in the upper left panel in Figure 8 and the
third column in the lower right panel in Figure 8). However, changing the rheological models of the
Pro- or Retro-terranes seems to have less impact on the deformation mode of the continental collisional
system, according to the simulation results of models which are connected by several cross-shaped
solid lines with different colors in Figure 8. Thus, it is difficult to determine whether the horizontal
strength contrasts between terranes or the vertical strength variation of a single terrane plays the
dominant role in a multi-terrane collisional system. This is also the significance and necessity of our
study.
**4.2. Influences of Lithosphere Structure**
Lithospheric thickness is one of the critical factors that control its strength (Burov, 2011), and it can
strongly vary between tectonic regions (Pasyanos et al., 2014). In our models, we assume different
lithospheric thicknesses for the Pro-, Mid- and Retro-terrane to explore these effects. Complex effects
are seen. When changing the lithospheric thicknesses of the Mid-terrane, or of all three terranes,
remarkable variations in lithosphere deformation appear in cases 1 and 2, but smaller variations are
seen for cases 3 and 4 (Figure 9). Cases 1 and 2 assume a Jelly sandwich rheology for the Pro-, Mid-
and Retro-terrane, so the strength of the lithospheric mantle of three terranes is comparable. Strength
variations produced by differences in lithospheric thickness may alter the relative strength of the three
terranes, resulting in distinct lithospheric deformations. For example, if the Pro- and Mid-terranes have
same lithosphere thickness, deformation mode in Case 2 would change from subduction to thickening
(subplot 3 vs. subplot 8 in Figure 9); if the Mid-terrane is thickest or the three terranes have same
thickness of lithosphere, deformation mode in Case 1 would change from collision to replacement
(subplot 2 vs. subplot 17 and 22 in Figure 9), and the polarity of the subduction of Pro-terrane's
lithospheric mantle would be reversed in Case 2 (subplot 3 vs. subplot 18 and 23 in Figure 9). Instead,
in cases 3 and 4, the Pro-, Mid- and Retro-terrane have two regions with stronger Jelly-sandwich-like
rheological structures and one with a weaker Crème brûlée structure, and deformation preferentially
concentrates in the weaker terrane. In comparison to the large strength difference implied for the
lithospheric mantle between the Crème brûlée and Jelly Sandwich rheological models, the strength
variations associated with the differences in lithosphere thickness are relatively small. Therefore,
changing the thicknesses of the lithosphere has much smaller effects on the lithosphere deformation, as
seen in cases 3 and 4 (also see the subplots in $3^{rd}$ and $4^{rd}$ rows of Figure 9).
In addition, the weak zones that suture two terranes are generally preserved during continental
amalgamation (Burker et al., 1977; Vink et al., 1984; Yin and Harrison, 2000). These local pre-existing
lithosphere weaknesses would be preferentially activated if the continental lithosphere were subjected
to compression, and could play a key role in concentrating deformation, adjusting deformation
sequences, and inducing lithosphere subduction (Sokoutis and Willingshofer, 2011; Heron et al., 2016;
Chen et al., 2020; Xie et al., 2021, 2023). Comparing the simulation results of models with and without
weak zone, we find that a weak zone will facilitate lithosphere subduction in earlier stages of model
evolution, resulting in more diverse lithosphere deformation patterns during the later stage (Figure 10).
**4.3. Implications for the Tectonics of East Asia**
**4.3.1. Lithosphere Collision beneath the Eastern Tien Shan**
The eastern Tien Shan is an ideal region to study the deformation patterns linked to long-term
lithosphere collision (Figure 11a). The eastern Tien Shan is bounded by the Tarim Basin to the south,
and the Junggar Basin to the north. It is composed of a series of former island arcs and small
continental blocks that amalgamated during the late Paleozoic (Han and Zhao, 2017). The lithosphere
of the eastern Tien Shan is weaker and thinner in comparison to its neighboring Tarim Basin and
Junggar Basin (Kumar et al., 2005; Lei and Zhao et al., 2007; Zhang et al., 2013; Deng and Tesauro,
2016). At ~20 – 25 Ma, the eastern Tien Shan became a reactivated orogeny in response to ongoing
India-Asia collision (Yin et al., 1998). Compression linked to the India-Asia collision induced the
Tarim lithosphere to underthrust northward (Xu et al., 2002; Guo et al., 2006; Lei and Zhao et al., 2007;
Lü et al., 2019; Hapaer et al., 2022; Sun et al., 2022). In the northern part of the eastern Tien Shan,
significant high-velocity anomalies and Moho overlap are also imaged, which are conventionally
explained as being due to the southward underthrusting of the Junggar lithosphere (Xu et al., 2002;
Guo et al., 2006; Li et al., 2016; Lü et al., 2019). High-velocity anomalies in the Tarim and Junggar
lithosphere appear to connect beneath the eastern Tien Shan, suggesting the lithosphere of the Tarim
and Junggar Basins has converged and collided together in this region (Figure 11b; Lü et al., 2019).
Bidirectional underthrusting of the Tarim and Junggar lithosphere leads to intense crustal shortening
and thrust faults on both flanks over the adjacent basins, as well as attendant fold and reverse fault
zones along the range fronts (Yin et al., 1998; Wang et al., 2004).
**4.3.2. Lithosphere Thickening and Delamination in the Tibetan Plateau**
The deformation pattern arising from lithosphere thickening and delamination has been applied to the
Tibetan Plateau (Figure 11c). Tibetan lithosphere may have been significantly weakened by hydration,
metasomatism, and partial melting of the lithospheric mantle during a series of oceanic closure and
terrane accretion events before the India-Asia collision (Yin and Harrison, 2000; Zhang et al., 2014;
Ma et al., 2021). It was then pushed northward by the Indian craton and was blocked by the
Tarim/Qaidam craton during India-Asia collision, leading to double crustal thickness (Zhao and
Morgan, 1985; Zhang et al., 2011). The lithosphere beneath the Tibetan Plateau does not thicken
significantly like its crust, especially beneath northern Tibet (Owens and Zandt, 1997; Tunini et al.,
2016). Numerous observations instead suggest that the Tibetan lithosphere has been detached from the
crust and has sunk into deeper mantle, consistent with the presence of high-velocity regions in the deep
mantle in western, southern and southeastern Tibet (Li et al., 2008; Chen et al., 2017; Feng et al., 2021).
A significant depression of the 660-km discontinuity beneath the Himalaya terrane and the uplift of
410-km discontinuity in western Tibet have also attributed to the presence of delaminated Tibetan
lithosphere (Wu et al., 2022). In northern Tibet, anomalously high temperatures are assumed to be
linked to a region of inefficient $S_n$ propagation indicating a thin or absent lithospheric mantle lid in this
region, while a remarkable low-velocity zone in the mantle and ultra-potassic volcanics also suggest
lithosphere thinning (Barazangi and Ni, 1982; Turner et al., 1996; Owens and Zandt, 1997; Guo et al.,
2006; Liang et al., 2012; Tunini et al., 2016). After lithosphere thinning commenced in the Miocene,
the Tibetan Plateau rapidly grew outwards (Lu et al., 2018 and references therein; Molnar et al., 1993;
Xie et al., 2023).

**4.3.3. Lithosphere Subduction in Southeastern China**

An example of intracontinental subduction is the Early Paleozoic Orogen in Southeastern China which
appears to have not been preceded by oceanic subduction (Figure 11d; Faure et al., 2009). The
northeasterly trending Early Paleozoic Orogen of Southeastern China is located on the Wuyi-Yunkai
Fold Belt which welds the Cathaysian Block to the south and the Yangtze Block to the north. Two
groups of models of collisional belt (Guo et al., 1989; Hsü et al., 1990) and intercontinental orogen
(Faure et al., 2009; Charvet et al., 2010; Li et al., 2010) have been proposed to explain the Early
Paleozoic Orogen in Southeastern China. Arguments against it being a collisional orogenic belt are its
lack of preserved ophiolites, a magmatic arc, subduction complexes, and high-pressure metamorphism;
instead, structural, metamorphic, and sedimentary elements indicate that this orogen was an
intracontinental orogen controlled by the northward subduction of Cathaysian Block (Faure et al., 2009;
Charvet et al., 2010; Li et al., 2010). At ~465 Ma, the Cathaysian Block underthrust beneath the
Yangtze Block along the Jiangshan–Shaoxing Fault, in which the lithospheric weaknesses inherited
from previous tectonic event of Nanhua rift at 800 – 850 Ma played an important role (Faure et al.,
2009; Charvet et al., 2010). During continental subduction, north-south horizontal shortening is
accommodated by ductile decollement zones in the Cathaysian Block, causeing remarkable
south-directed crustal folding and thrusting structures related to both thin-skinned and thick-skinned
tectonics in the Wuyishan proper and its southern border, and north-directed structures to the west of
Ganjiang Fault and north of Jiangshan–Shaoxing Fault, where only thin-skinned tectonics is visible (Li
et al., 1998; Shu et al., 1999; Charvet et al., 2010). The tectonic development of the Early Paleozoic
Orogen in Southeastern China appears similar to the deformation mode of lithosphere subduction
(Figure 4, 10a and 10c).
So far, we have yet to find a suitable region to apply the model deformation pattern of lithosphere
replacement. In this deformation pattern, crustal deformation and topographic evolution are similar to
those in the deformation pattern of lithosphere collision (Figures 7a and 7c). Thus, it is not easy to
identify this pattern by geological and geophysical techniques when the replaced and original
continental lithosphere has similar properties. Improved imaging observations with better resolution
may allow this deformation pattern to be identified in the future.

## 5. Model Limitations

Although we can obtain four deformation modes of continental lithosphere by changing the rheologies
of different terranes in a collisional model, we must keep in mind that our results are based on some
simplifications and assumptions, which may affect the model results. For example, in our model three
terranes are directly collaged together, but in nature different terranes are often connected through weak
sutures which may preferentially deform when they are subjected to compression (Burker et al., 1977;
Yin and Harrison, 2000). These local pre-existing weak zones have non-negligible influences on
lithospheric deformation, and their role were widely discussed in previous studies (Sokoutis and
Willingshofer, 2011; Heron et al., 2016; Chen et al., 2020 ; Xie et al., 2021, 2023). We also discussed
the effects of local pre-existing weak zones in Section 4.2. In addition, lithosphere thicknesses of the
Pro-, Mid- and Retro-terranes are chosen arbitrary in our models, but they also have important
influences on lithosphere deformation (Figure 9). Some studies suggest that differences in crustal
strength will also cause different lithospheric deformation (Faccenda et al., 2008; Vogt et al., 2017,
2018), but the three terranes are set same crustal structure in our model for the aim of simplification. As
well, some studies believe that the convergence rate will greatly affect the deformation of orogenic
belts (Chen et al., 2016; Vogt et al., 2017), but in this study, the impact of the convergence rate is not
been discussed.

## 6. Conclusions

The continental lithosphere is likely to have strong lateral variations in its strength. We explored 2-D
numerical models that contain three diverse types of continental terranes to study the responses of
continental terranes with different strengths to compression. Four rheological models were respectively
applied to each of the Pro-, Mid- and Retro-terranes, and simulation results can be grouped into four
distinct deformation styles:   lithosphere collision, subduction, thickening and delamination, and
replacement. These deformation styles arise from the rheological contrasts between the terranes: (1)
when the middle terrane is the weakest, its lithosphere is easily extruded, which leads to lithosphere
collision between its two bounding terranes; (2) when the middle terrane has a strong lower crust, while
the lower crust of a bounding terrane is weak, then subduction of the lithosphere of the bounding
terrane will occur; (3) when a bounding terrane is the weakest, its lithosphere would tend to be
thickened by lateral compression, followed by lithosphere delamination due to the resulting
density/gravitational instability; (4) when a bounding terrane has a strong lower crust and weak
lithospheric mantle, while the middle terrane has a weak lower crust and strong lithospheric mantle,
then lithosphere replacement will occur. These simulation patterns are seen in observed deformation
patterns and structures in the eastern Tien Shan, and the Tibetan Plateau, the Early Paleozoic Orogen of
Southeastern China, suggesting they are likely to be naturally occurring modes of intracontinental
orogenesis.
*Code availability*
Requests for the numerical code I2VIS should be sent to the main developer
(taras.gerya@erdw.ethz.ch).
*Data availability*
Numerical modeling data and the model evolution animations of Cases 1 – 4 are all provided in
Zenodo (https://doi.org/10.5281/zenodo.8354366 and https://doi.org/10.5281/zenodo.10731981).

*Author contribution*: **Conceptualization:** Yongshun John Chen; **Methodology:** Lin Chen, Renxian Xie; **Investigation:** Renxian Xie; **Formal analysis:** Renxian Xie, Lin Chen; **Visualization:** Renxian Xie, Jason P. Morgan; **Writing – original draft preparation:** Renxian Xie; **Funding acquisition:** Yongshun John Chen, Lin Chen, Renxian Xie.

*Competing interests*: The authors declare that they have no conflict of interest.

*Disclaimer. Publisher's note*: Copernicus Publications remains neutral with regard to jurisdictional claims in published maps and institutional affiliations.

*Acknowledgments*

The authors sincerely thank Prof. Taras Gerya for providing the I2VIS package and his long-lasting guidance on our geodynamic modeling. We are also grateful to the two anonymous reviewers for their insightful comments, which greatly improved the presentation of the paper. The authors acknowledge that figures of simulation results were prepared with the Generic Mapping Tools (GMT, http://www.soest.hawaii.edu/gmt/), and the color bar of batlow used in the figures of viscosity field is from Crameri et al. (2018). All models were performed on the TianHe-1A system at the National Supercomputer Center in Tianjin.

*Financial support*: This study was supported by the National Natural Science Foundation of China (Grants U1901602), the National Key R&D Program of China (2022YFF0800800) and National Natural Science Foundation of China (42374076).

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

**Figures and captions**

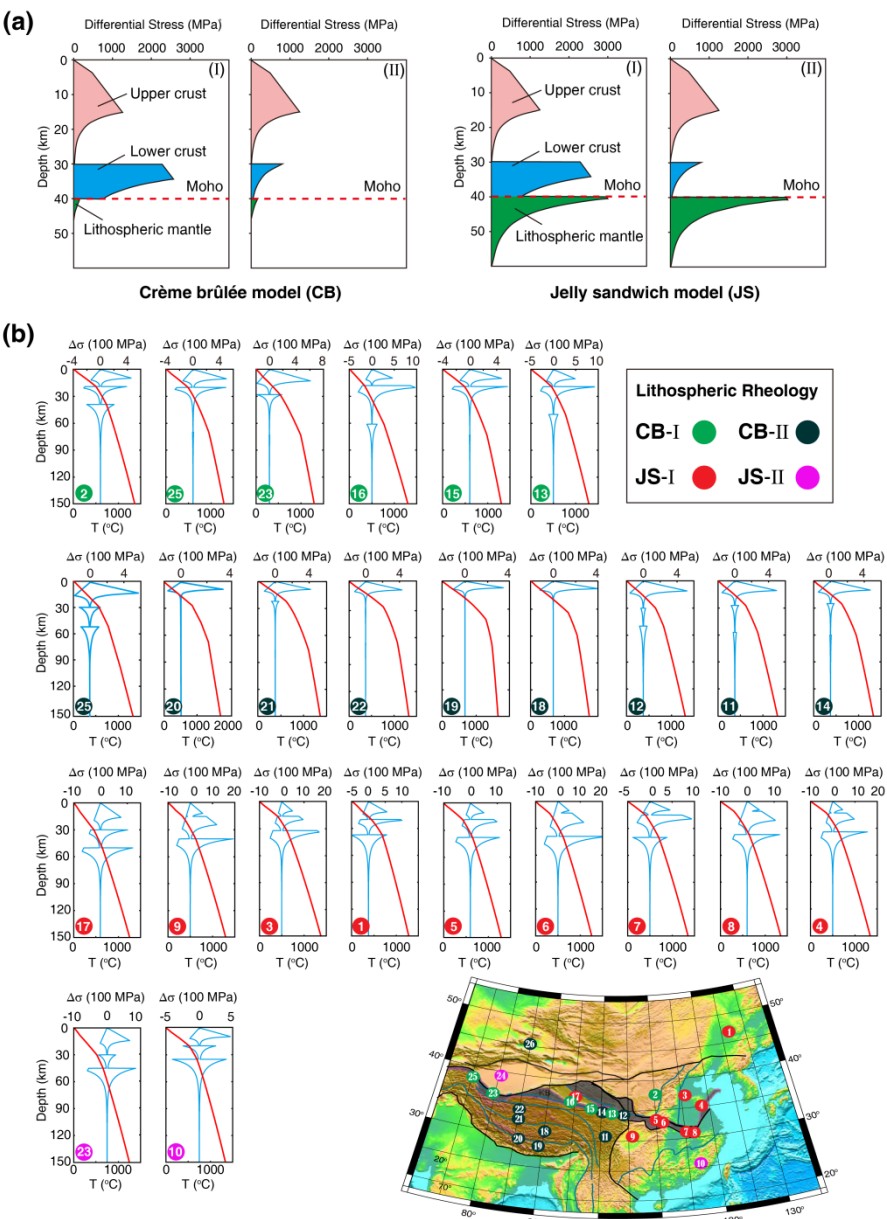


**Figure 1. Four rheological models of continental lithosphere.** (a) Crème brûlée model (CB) and Jelly sandwich model (JS). The two rheological models can be further subdivided into CB-I, CB-II, JS-I, and JS-II according to the strength of the lower crust (modified from Jackson, 2002). (b) Observations of four distinct lithosphere rheological structures implied for East Asia (modified from Zhang et al., 2013). Locations of strength profiles are pointed out by dots with numbers in the topography map. Dots filled with different colors indicate different models of lithospheric rheology. These strength profiles are calculated based on observed geothermal structure and lithospheric structure, and assumed that compositions of the upper and lower crust and lithospheric mantle are wet quartzite, undried granulite and dry olivine, respectively. Variations of temperature and lithospheric compositions

lead to a diverse suite of strength profiles vs. depth.

701

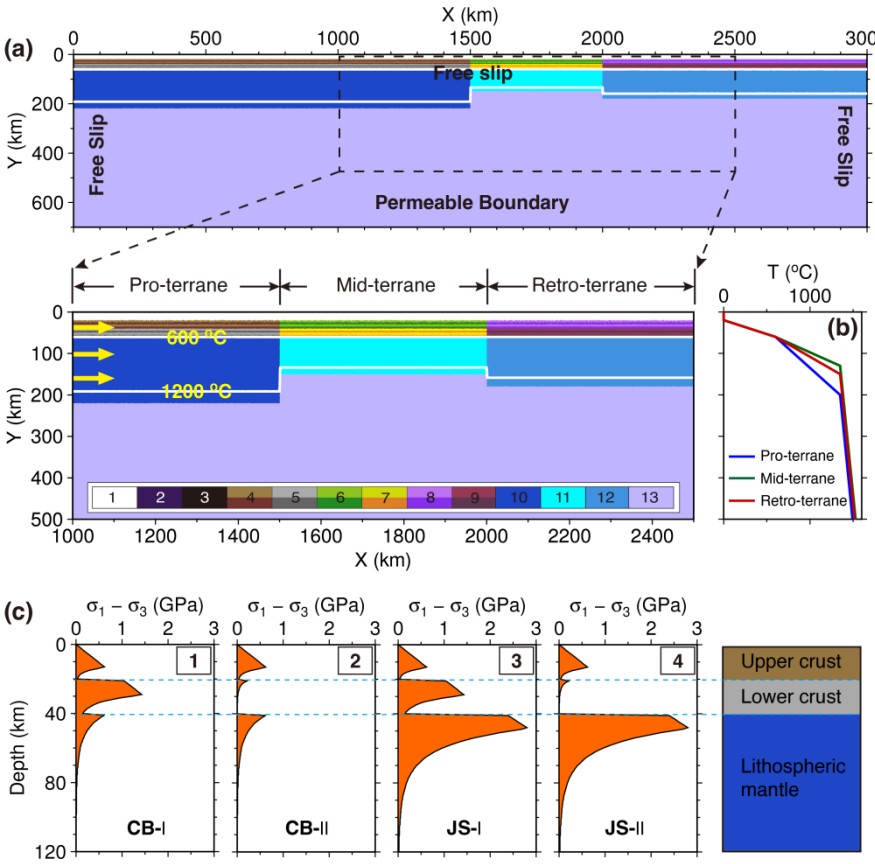

**Figure 2. Initial model setup.** (a) Initial model configuration. The model size is 3000 km × 700 km, and size of study region is 1500 km × 500 km. Three continental terranes of the Pro-, Mid- and Retro-terranes are contained in the numerical model, and they are 200 km, 130 km, and 160 km thick, respectively. White lines are isotherms with an interval of 600 ℃. Yellow arrows indicate the convergence rate of 20 mm/yr. Colored grids: 1 – sticky air; 2 – sediments; 3 – weak zone; 4, 6, 8 – the upper crust of the Pro-, Mid- and Retro-terranes, respectively; 5, 7, 9 – the lower crust of the Pro-, Mid- and Retro-terranes, respectively; 10, 11, 12 –lithospheric mantle of the Pro-, Mid- and Retro-terranes, respectively; 13 – asthenosphere. (b) Initial temperature structure for the three terranes. The Pro- and Mid-terranes respectively have a coldest and warmest lithospheric mantle due to their differences of lithosphere thicknesses. (c) Four rheological models with contrasting lithospheric strength profiles. These are derived from different strength scaling factor (*S*) combinations for the upper crust, lower crust, and lithospheric mantle (Table S1). Strength profiles are calculated based on the Pro-terrane's initial lithospheric structure, composition, and temperature field. The prescribed strain rate is $1 \times 10^{-14}$ s$^{-1}$. CB-I and CB-II, the crème brûlée model with strong and weak lower crust, respectively; JS-I and JS-II, the jelly sandwich model with strong and weak lower crust, respectively.

717

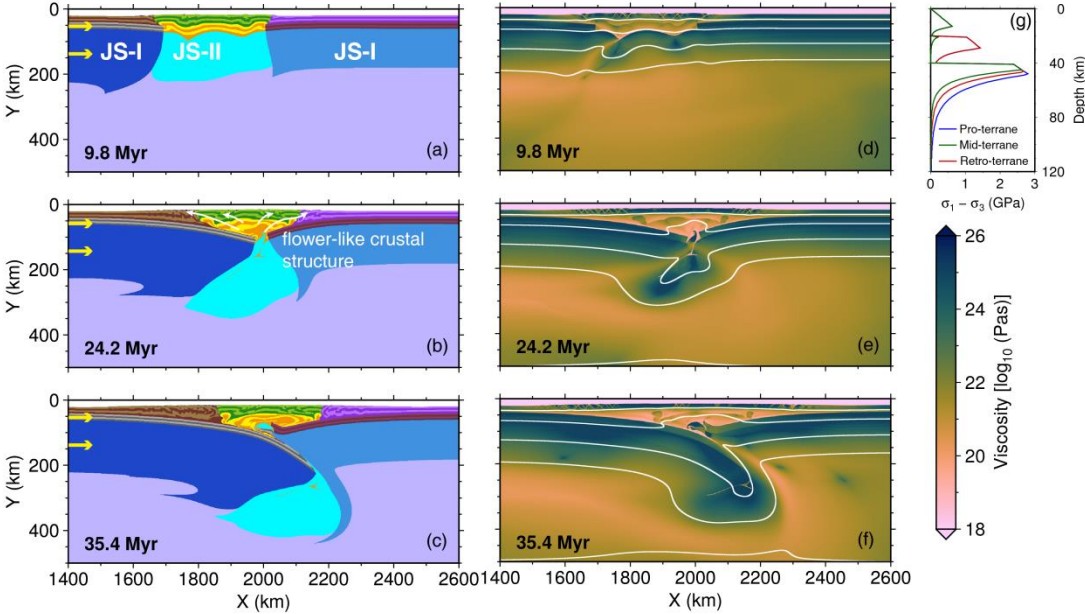

718

**Figure 3. Collision of the lithospheres of the Pro- and Retro-terranes.** Rheological models for the Pro-, Mid- and Retro-terranes are JS-I, JS-II, and JS-I, respectively, as shown in (g). The left panel shows compositional fields at 9.8 Myr, 24.2 Myr, and 35.4 Myr, respectively. Yellow arrows indicate the convergence rate. The right panel shows the corresponding viscosities. White lines are isotherms with an interval of 300 °C.


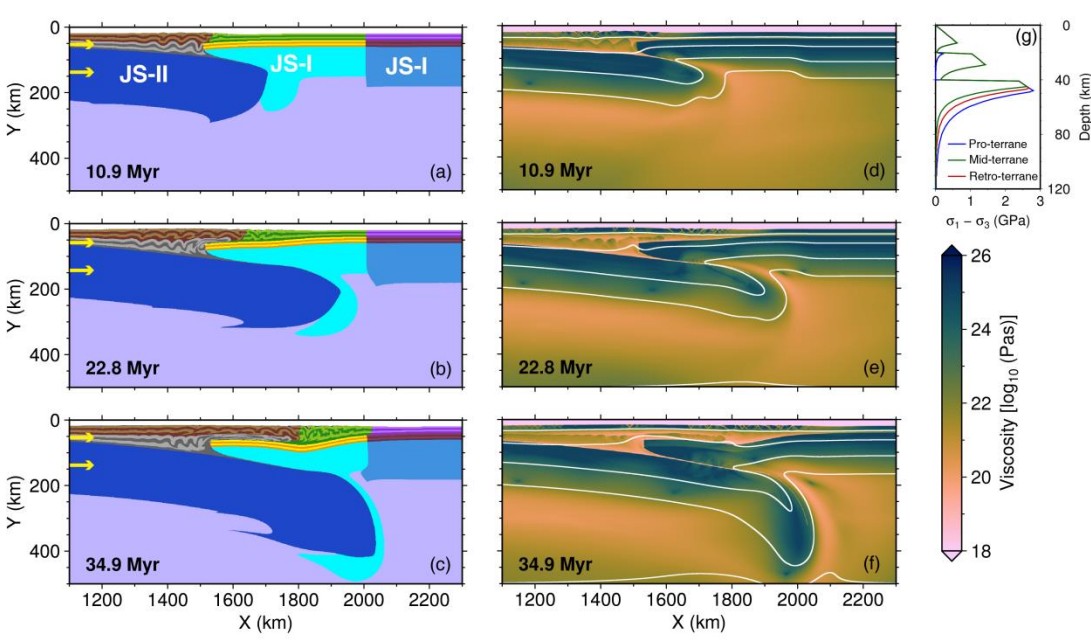


**Figure 4. Subduction of the lithosphere of the Pro-terrane.** Rheological models for the Pro-, Mid- and Retro-terranes are JS-II, JS-I, and JS-I, respectively. See Figure 3 for plotting conventions.


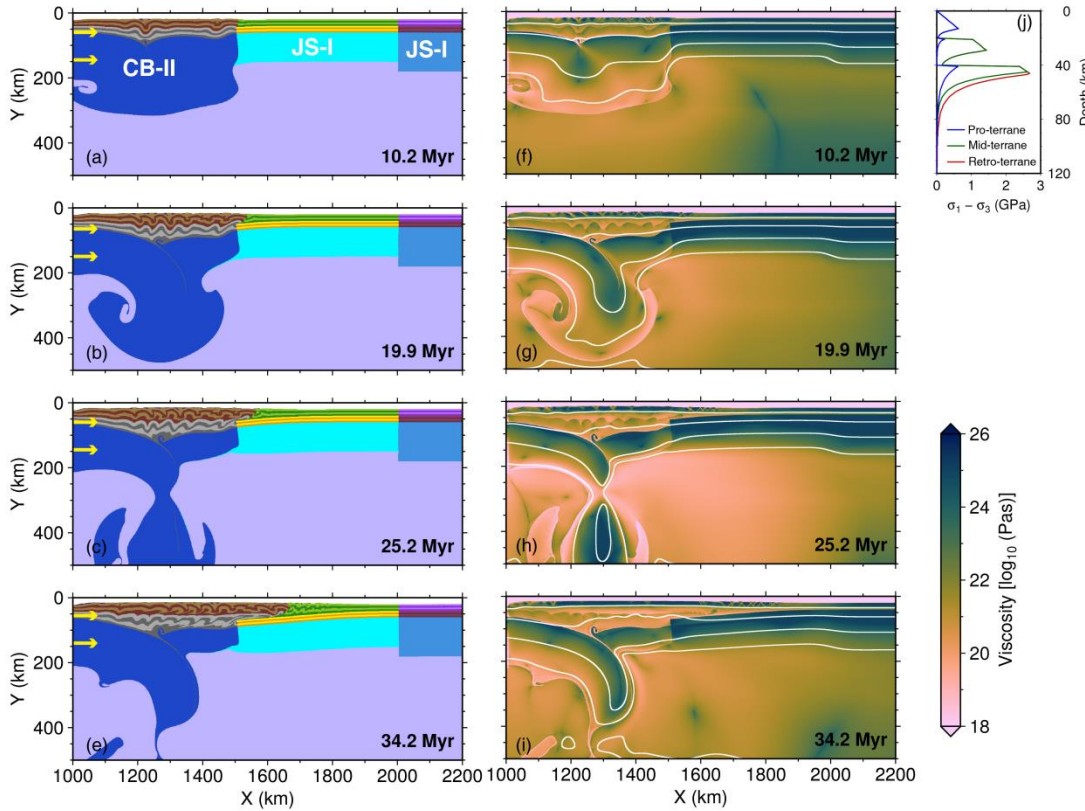


**Figure 5. Thickening and delamination of the lithosphere of the Pro-terrane.** Rheological models for the Pro-,

Mid- and Retro-terranes are CB-II, JS-I, and JS-I, respectively. See Figure 3 for plotting conventions.


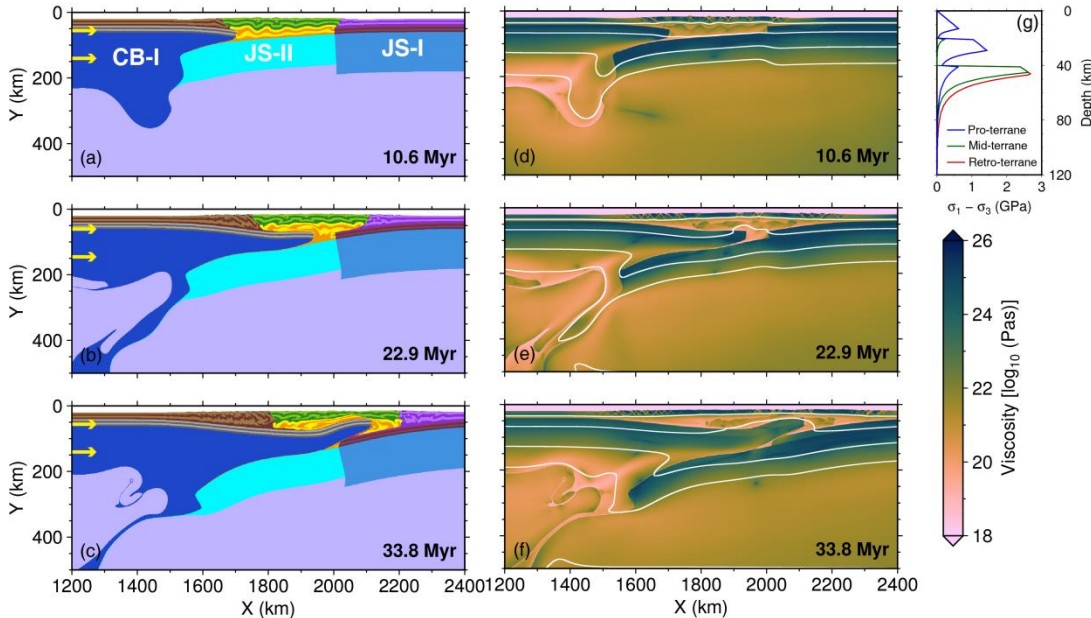


**Figure 6. Replacement of the lithosphere of the Pro-terrane.** Rheological models for the Pro-, Mid- and

Retro-terranes are CB-I, JS-II, and JS-I, respectively. See Figure 3 for plotting conventions.

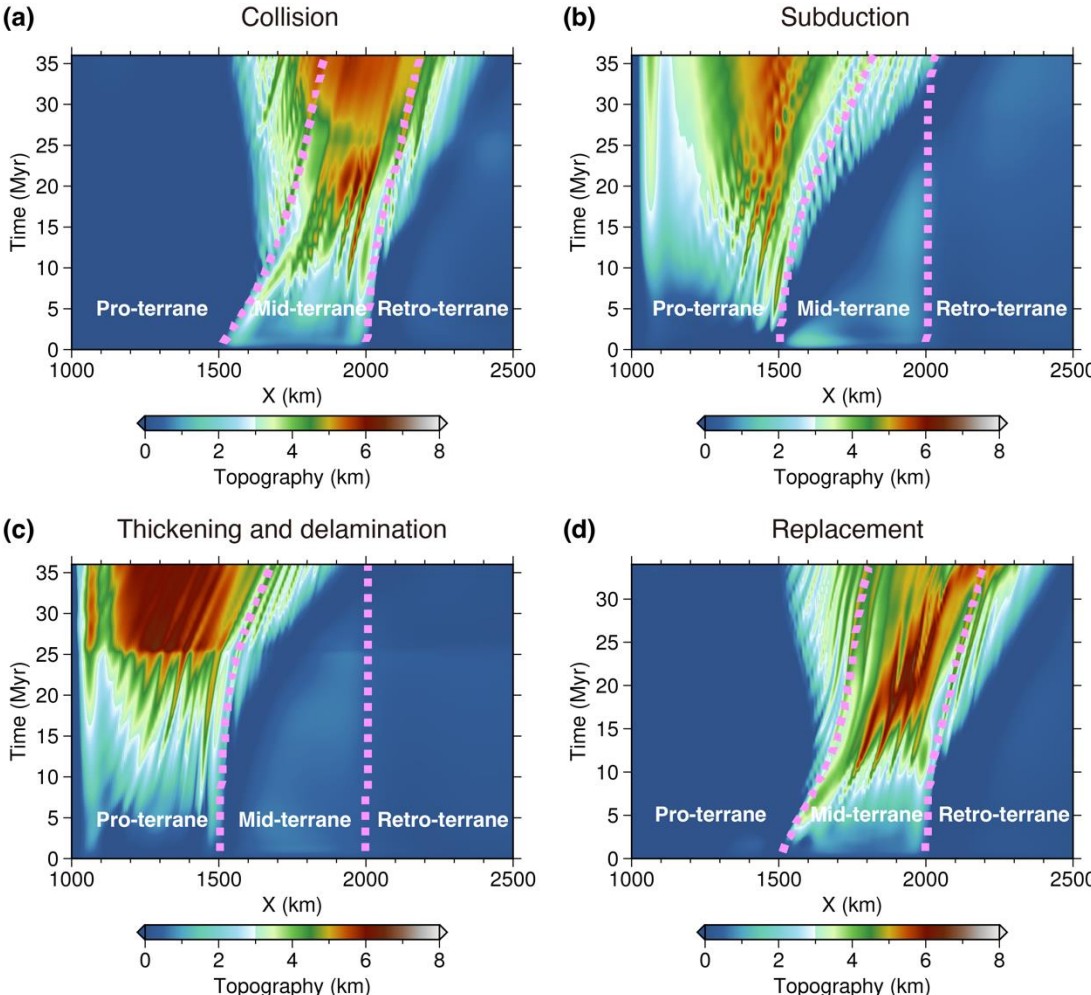


**Figure 7. Evolution of surface relief for the different deformation styles.** The purple dashed lines indicate the

boundaries between terranes. (a) – (d) Surface relief associated with the deformation patterns of lithosphere

collision, subduction, thickening and delamination, and replacement, respectively.


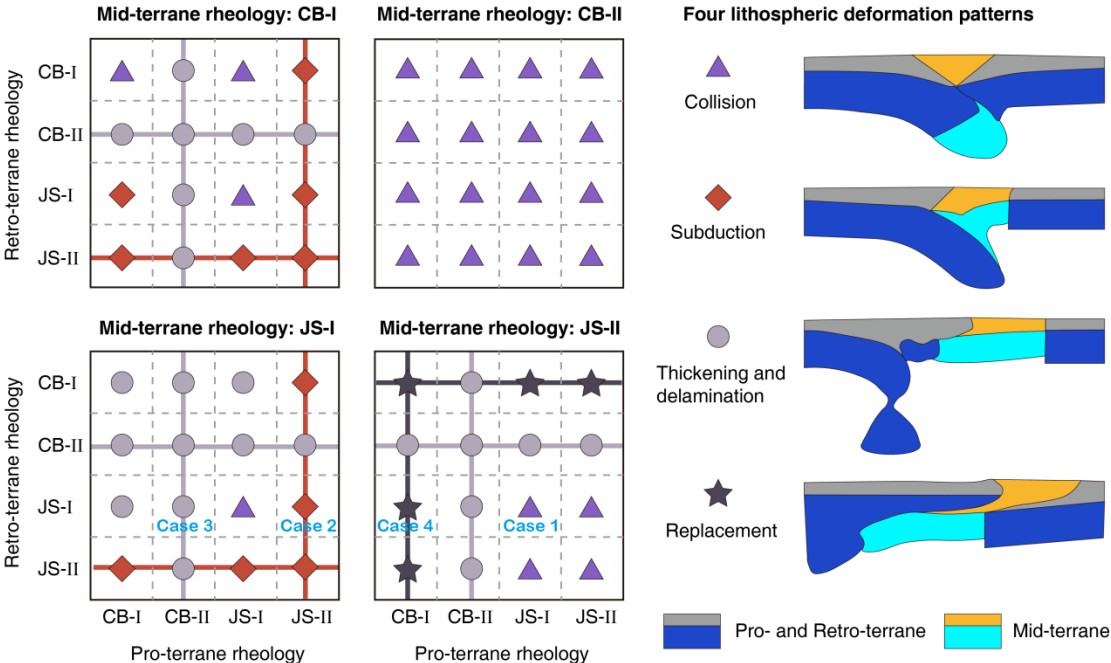


**Figure 8. Four styles of lithosphere deformation patterns.** Symbols with colors indicate different deformation

patterns of the lithosphere. Cases 1 – 4 are the selected models chosen to illustrate details of these modes of

compressional evolution.

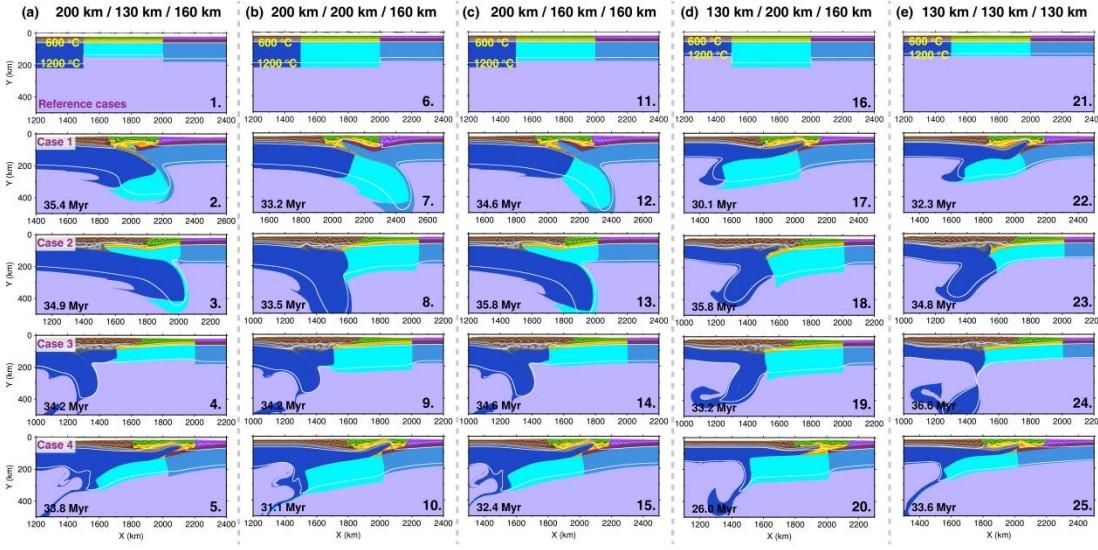

746

**Figure 9. Effects of lithosphere thicknesses of various terranes.** (a) – (e) Final simulation results of models with

the different lithosphere thicknesses of the Pro-, Mid-, and Retro-terranes, respectively. (a) Final simulation

results of reference cases. Rheological models of the Pro-, Mid- and Retro-terranes in 2 – 4 rows are same with

those in Cases 1 – 4, respectively.

751

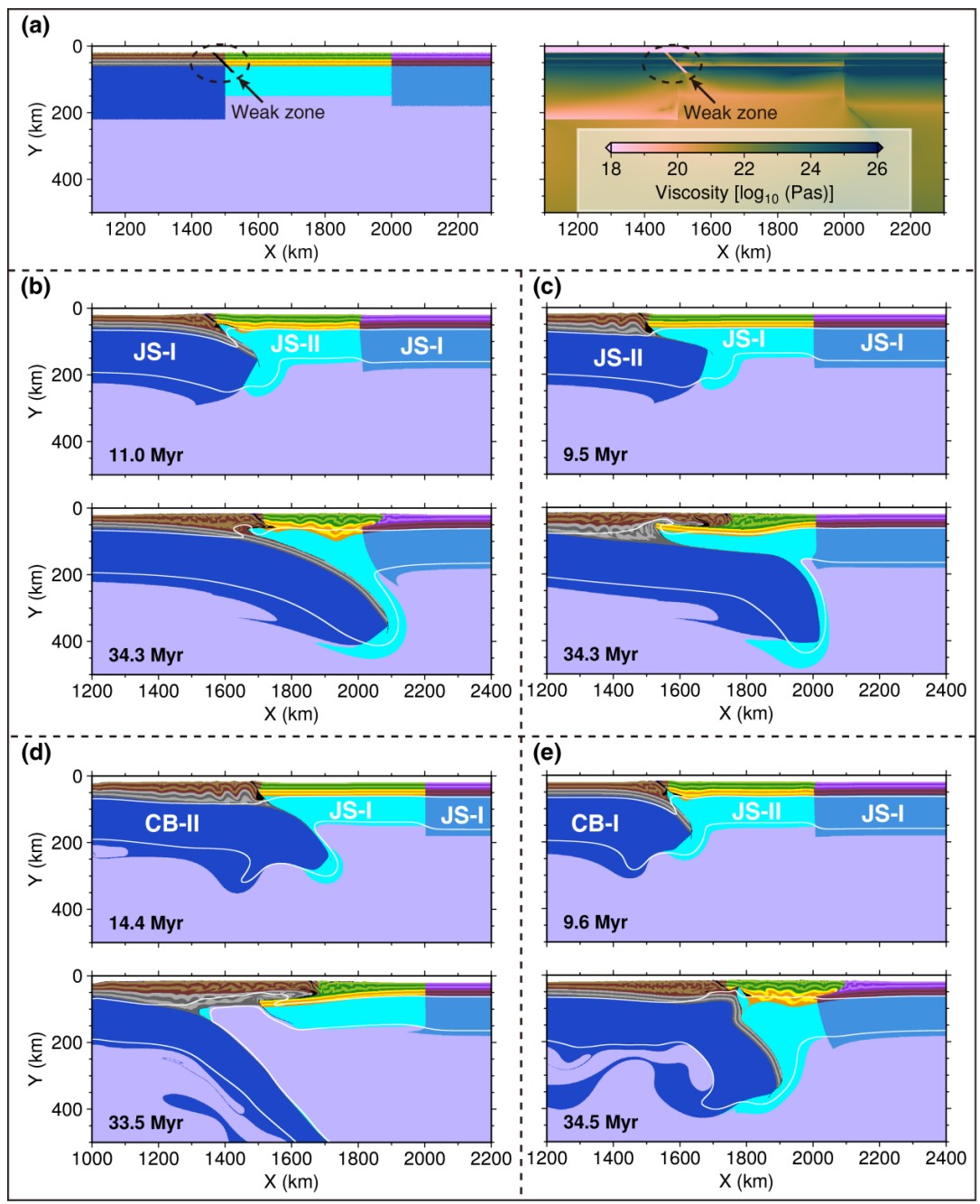

**Figure 10. Effects of local weak zone on lithosphere deformation.** (a) Details about the weak zone. (b) – (e)

Final simulation results of models corresponding to Cases 1 – 4, respectively.

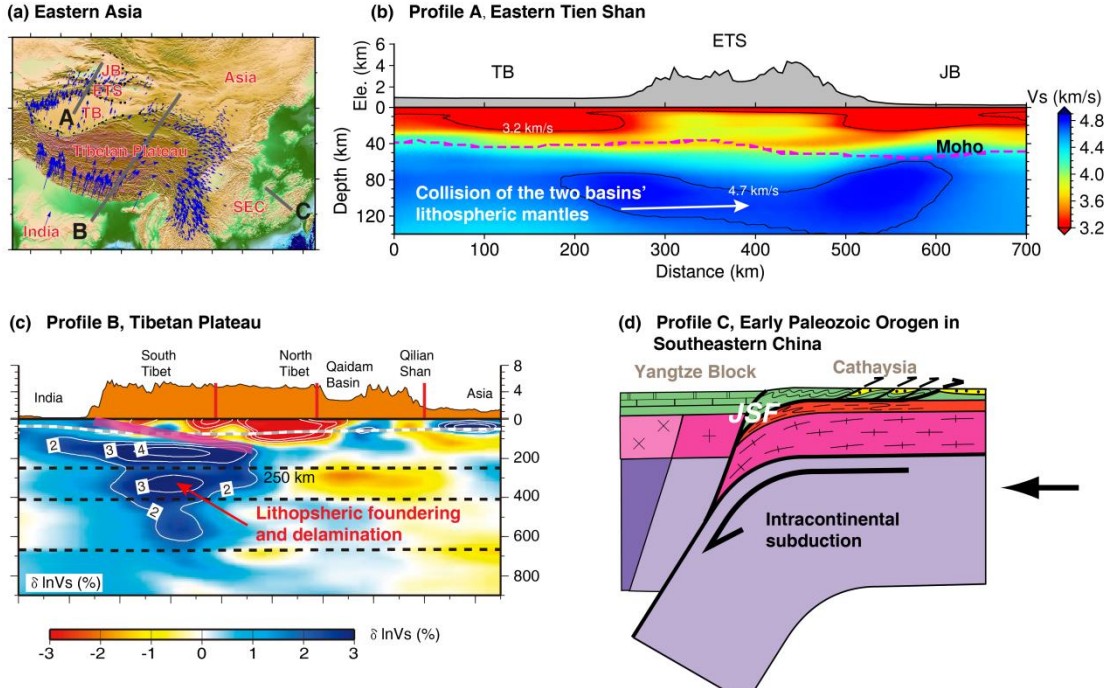


**Figure 11. Implications of simulation results to East Asia.** (a) Topography of East Asia. The three gray lines
point out the locations of lithosphere profiles in (b), (c) and (d). TB, Tarim Basin; ETS, eastern Tien Shan; JB,
Junggar Basin; SEC, Southeastern China. (b) Collision of the lithospheric mantle of Tarim Basin and Junggar
Basin beneath the eastern Tien Shan (modified from Lü et al., 2019). (c) Lithospheric founding and delamination
in the Tibetan Plateau (modified from Chen et al., 2017). (d) Intracontinental subduction in the Early Paleozoic
Orogen in southeastern China (modified from Faure et al., 2009). JSF, Jiangshan–Shaoxing Fault.
**Table 1. Flow laws and material properties for different lithospheric layers.** $\rho_0$ is the initial density; it evolves
with time as $\rho = \rho_0 \left(1 - \alpha\left(T - T_0\right)\right)\left(1 + \beta\left(P - P_0\right)\right)$, where $T_0 = 20\,^{\circ}\text{C}$, $P_0 = 10^5$ MPa. Flow law: qtz. = quartzite, Plag.
= plagioclase, ol. = olivine.

| Material properties | Sediment | Upper crust | Lower crust | Lithospheric mantle | Asthenosphere |
|---|---|---|---|---|---|
| $\rho_0$ (kg/m$^3$) | 2600 | 2700 | 2800 | 3300 | 3300 |
| Flow laws | Wet qtz. | Wet qtz. | Plag. | Dry ol. | Dry ol. |
| $1/A_D$ (Pa$^n$ s) | $1.97 \times 10^{17}$ | $1.97 \times 10^{17}$ | $4.80 \times 10^{22}$ | $3.98 \times 10^{16}$ | $3.98 \times 10^{16}$ |
| $n$ | 2.3 | 2.3 | 3.2 | 3.5 | 3.5 |
| $E_a$ (KJ/mol) | 154 | 154 | 238 | 532 | 532 |
| $V_a$ (J/bar) | 0.8 | 0.8 | 1.2 | 1.2 | 1.2 |
| $\phi = \sin(\varphi)$ | 0.2 – 0.1 | 0.3 – 0.1 | 0.3 – 0.1 | 0.6 – 0.4 | 0.6 – 0.3 |
| $C$ (Pa) | $1 \times 10^{7-6}$ | $1 \times 10^{7-6}$ | $1 \times 10^{7-6}$ | $1 \times 10^{7-6}$ | $1 \times 10^{7-6}$ |
| $H_r$ (uW/m$^3$) | 2.0 | 1.5 | 0.5 | 0.022 | 0.022 |
| $C_p$ (J/kg K) | 1000 | 1000 | 1000 | 1000 | 1000 |
| $\alpha$ (1/K) | $3 \times 10^{-5}$ | $3 \times 10^{-5}$ | $3 \times 10^{-5}$ | $3 \times 10^{-5}$ | $3 \times 10^{-5}$ |
| $\beta$ (1/MPa) | $1 \times 10^{-5}$ | $1 \times 10^{-5}$ | $1 \times 10^{-5}$ | $1 \times 10^{-5}$ | $1 \times 10^{-5}$ |
| $k$ (W/m/K) | $0.64 + 807/(T+77)$ | $0.64 + 807/(T+77)$ | $1.18 + 474/(T+77)$ | $[0.73 + 1293/(T+77)] \times (1 + 0.00004P)$ | $[0.73 + 1293/(T+77)] \times (1 + 0.00004)$ |
