# Peer review of "Various lithospheric deformation patterns derived from"

_EGUsphere, 2023_

## Author Comment (AC2)

Figures in the Responses to Reviewer 1's comments:

[Figure]

Figure R1. Evolution of the composition and temperature fields of model without convergence rate. In this model, rheology models of the Pro-, Mid- and Retro-terranes are JS-I, JS-II and JS-I which are same to the Case 1. White lines are isotherms with an interval of 200° C.

[Figure]

Figure R2. Strength of the Pro-, Mid- and Retro-terranes in model of JS-I/JS-II/JS-I (case 1).

[Figure]

Figure R3. Crustal deformation and topography uplift of deformation mode of lithosphere subduction. The upper, middle and lower panels are topography, component and strain rate results, respectively.

[Figure]

Figure R4. Delamination of the Pro-terrane's lithospheric mantle. The left, middle and right panels were the component results of models of CB-II/CB-I/CB-I, CB-II/CB-I/JS-I, and CB-II/CB-I/JS-II, respectively.

---

## Author Response (AR1)

**Comments of** "*Various lithospheric de formation patterns derived from rheological contrasts between continental terranes: Insights from 2-D numerical simulations*"
* * *
**Comments from Referee #1:**

The manuscript titled "*Various lithospheric deformation patterns derived from rheological contrasts between continental terranes: Insights from 2-D numerical simulations*" by Renxian et al. addresses the topic of continental collision-dynamics in the presence of multiple terranes using two-dimensional thermo-mechanical numerical experiments. The authors aim to explore the effects of rheological differences between the colliding terranes on the dynamics of collision and the style of the developing orogen. They use 4 different base-rheologies, slightly modulated by the different lithospheric thicknesses assigned for the 3 terranes. The resulting parameter-space is mapped out in full and 4 different "cases" of first-order deformation patterns are identified that vary systematically with the rheological settings. Finally, the authors try to match the observed "cases" to natural systems from around the globe.

The manuscript is mature and generally proportional and well written. The figures are clear, neat and all contribute substantially to the arguments laid out. The supplementary figures were also very useful in assessing the results. I am missing a set of supplementary animations and maybe an additional set of supplementary figures that complement the current ones but present viscosity structure. These would allow for a better understanding of what actually happens in each experiment. I recognize that the set is fairly large and it might be impractical to include an animation for all of the models but one for each type-example should be doable.

In general, I do have a number of concerns about the study, and especially about the classification and interpretation of the results. I would argue that the interpretation needs at least a strong second-look. More and stronger arguments are needed to be provided about the validity of the classification-system and why each experiment has been put in the class they are in. I will only detail my main concerns below but will also provide an annotated pdf file with all my various comments and remarks large and small.

Responses: We sincerely thanks for your suggestions on improving our manuscript. We added final viscosities results of all models as supplementary Figure S2 in the supplements file, and supplemented four animations showing the evolution of topography, component and viscosities for each type-example, respectively (please see *Data availability*). We also summarized the main deformation features of each modes before describe the representative simulation results, so that we can easily distinguish and classify each simulation results.
* * *
**1.** The choice of the different lithosphere-thicknesses for the pro-, mid- and retro blocks seems a bit arbitrary to me. Is it reasonable that all three terranes have the

same crustal structure, even though the thicknesses of the underlying mantle-lithospheres vary so greatly (between 50 km and 120 km)? It is also unclear for me from the discussion of the methods whether the lithospheric blocks are in thermal equilibrium at the start of the experiments or not, and if they are not whether it has an effect on their evolution. I would argue that all three terranes cannot be stable with the prescribed setup, leading to Moho-temperatures that evolve over time regardless of whether they undergo deformation or not.

The Authors did run a sensitivity test where they varied the lithospheric thicknesses of the three terranes for a representative example of each of the four identified deformation cases which is commendable but even there a setup where the mid-terrane is thicker then both neighbouring terranes is missing. I believe that the manuscript would benefit from at least a short discussion about why these lithospheric structures were chosen. Maybe a test where the representative models are set up so that the lithospheres are at thermal at equilibrium at the start of the model would also help. If the authors can convincingly argue that the equilibrium thermal structure is close enough that it would have no effect on the model evolutions that can of course also help.

Responses #1: We answered this comment from the following aspects:

(1) We set different thicknesses of the lithospheric mantle for the three terranes in our models to reflect the heterogeneous thickness of continental lithosphere (Pasyanos et al., 2014). Their thickness was chosen arbitrarily. We discussed the effects of lithospheric thickness on deformation in section 4.2 and also emphasized this in the Model Limitations. At the same time, for the purpose of simplification, we set the three terranes to the same crustal structure. We have rewritten this part in section 2. Please see Lines 150-156.

(2) The lithospheric blocks are not in thermal equilibrium at the start of the experiments. Simulation results of the model without convergence rate show that the temperature field would mildly evolve over time but the lithosphere has little deformation even if in a long-term period (Figure R1). Thus, we believe that the effects resulting from non-thermal equilibrium lithospheric blocks can be ignored. We stated this in Model setup. Please see Lines 174-176.

(3) We additionally tested 4 models in which the Mid-terrane is thicker than the surrounding terranes, and their simulation results were added as Figure 9d.

[Figure]

Figure R1. Evolution of the composition and temperature fields of model without convergence rate. In this model, rheology models of the Pro-, Mid- and Retro-terranes are JS-I, JS-II and JS-I which are same to the Case 1. White lines are isotherms with an interval of 200 °C.

**2.** The manuscript is also missing a limitations section. This is always useful to present in modelling studies but in this case, I would argue that it is doubly important when considering the results of the experiments that contain zones of inherited weaknesses representing paleo-sutures. In my opinion those models show that at least in some cases structural inheritance can override the deformation preferences that arise from the compositional rheological settings. This gives an important context when interpreting the results that – I think – should be spelled out more explicitly.

Responses #2: We added "Model Limitations" as section 5, in which we stressed the potential effects of pre-existing weak zones, lithosphere structure and convergence rate. Please see Lines 409-424.

**3.** Regarding the classification of the observed behaviours – and this is my main concern – it is not always clear to me what exact criteria were used to classify the individual experiments. I think the criteria are actually there at the end of each of the four sub-sections (3.1 to 3.4) describing the type-examples of the cases but I found the descriptions a bit muddled (especially in the first case). I think the paper might benefit from presenting these criteria together at around line 175. Furthermore, it is unclear to me why these criteria were chosen? Do they really mark fundamentally different behaviours?

I am asking these questions because a lot of the time, I could not tell from the

provided single snapshots what were the reasons for placing a particular experiment into the "thickening and delamination" or "collision" cases and not into the "subduction" case (see examples in the attached commented version of the manuscript in the comment around the lines 230-232). Is it purely because one bounding terrane remains largely undeformed in these experiments? I would argue that that fact alone does not make the observed behaviour delamination or collision as opposed to subduction.

Regarding the type-example shown for the "replacement" case, I would argue that the very weak bottom portion of the Pro-terrane mantle-lithosphere drips. It seems to be an order of a magnitude weaker than the underlying sub-lithospheric mantle, so this is not a huge surprise. In the meantime, the Mid- and Retro-terrane mantle-lithosphere under-thrusts the strong upper portion of the Pro-terrane mantle-lithosphere, initiating continental subduction.

Responses #3: We added the main deformation features of different deformation modes to distinguish them before describe their detailed evolutional processes. Please see Lines 185-193.
We also provided 4 videos as supplementary materials, by which we can observe the detailed evolutional processes of each of deformation modes

**4.** In the discussion, when setting out the reasons behind the observed behaviour of the "collision" cases, the authors argue that: "*When the Mid-terrane' mantle is weakest (typified by models in which the rheological model of the Mid-terrane is CB-II), it is easy for its mantle to be extruded, leading to collision between the lithospheric mantles of its surrounding Pro- and Retro-terranes.*"

However, there are models classified as "collisional" where the Mid-terrane mantle-lithosphere is strong but its lower-crust is weak (JS-II)? This means that authors' argument here explains a large portion the experiments classified as "Collision" – 19 out of 24 if I count the ones where the mid-terrane rheology is CB-I – but there are still 5 models where the mantle-lithosphere of the mid-terrane does not appear to be weaker than the mantle-lithosphere of the surrounding terranes. How come these models show collisional behaviour? As a side-note, the experiment presented as a type-example of this behaviour is one of the 5 models where the provided explanation does not seem to be valid.

Responses #4: The 5 models may be JS-I/JS-I/JS-I, JS-I/JS-II/JS-I (Case 1), JS-I/JS-II/JS-II, JS-II/JS-II/JS-I, and JS-II/JS-II/JS-II, in which the rheology models of the Pro-, Mid- and Retro-terranes are all JS. Even so, there are still differences in the strength of their lithospheric mantles owing to the differences in lithosphere thicknesses. The Mid-terrane's lithospheric mantle is weakest due to its thinnest lithosphere (the green line in Figure R2). Therefore, we say "*When the Mid-terrane's lithospheric mantle is weakest (…), it is easy for its mantle to be extruded out, leading to collision between the lithospheric mantles of its surrounding Pro- and Retro-terranes*" is suitable for all collision models.

[Figure]

Figure R2. Strength of the Pro-, Mid- and Retro-terranes in model of JS-I/JS-II/JS-I (case 1).

**5.** Regarding the natural examples: I am not an expert on any of these systems so I do not feel particularly competent at commenting on the appropriateness of the comparisons. However, I would like to point out, that there is an odd imbalance in how detailed the comparison is for the three cases where similarly behaving natural examples are discussed. In fact, only one of the systems is shown on a figure; there are three panels dedicated to the Eastern Tien Shan but not for the other two examples. I would suggest that the authors should include a cross-section of some sorts for the other examples as well.

Responses #5: We divided section 4.3 into three small parts, and applied the simulation results to the eastern Tien Shan, the Tibetan Plateau and the Early Paleozoic Orogen in Southeastern China, respectively. Please see section 4.3.1-4.3.3.

At the same time, we added the cross-sections of the Tibetan Plateau and the Early Paleozoic Orogen in Southeastern China. Please see Figure 11.

**6.** I have one additional question regarding the Eastern Tien Shan example: Based on the seismic tomography study of Lü et al. 2019, used and cited here, there is considerable variation in the deep behaviour of the Tarim and Junggar basins and the Tien Shan along an east-west transect (from subduction to "upwelling" to collision). In the Author's opinion, could this be attributed to changes in rheology or is this variation caused by something else?

Responses #6: We also noticed the considerable variation in the deep behavior in the Tien Shan along an east-west transect from collision to mantle upwelling to subduction (Lü et al. 2019). Mantle upwelling beneath the central Tien Shan may be caused by lithosphere underthrusting/subduction or delamination (Lü et al. 2019; Zhang et al., 2022). In fact, the lithospheric mantle of the western Tarim Basin was modified by mantle plume during the Mesozoic leading to it is stronger and more

buoyant (Xu et al., 2020), which is not conducive to subduction. So, why the Tarim lithosphere subducted northward beneath the central and western Tien Shan but collided with the lithosphere of the Junggar Basin? We guess it may be the following two reasons based on our simulation results:

(1) The lower crust in the central and western Tien Shan is stronger than that in the eastern Tien Shan. Our simulation results show that strong lower crust of the middle terrane can facilitate lithosphere subduction (Figure 8). If more ocean basins were trapped in the crust of the central and western Tien Shan than the eastern Tien Shan during the continental growth (Han and Zhao, 2017; Yang et al., 2022), then it is possible that it would have a stronger lower crust. However, distinguishing these trapped ocean basins needs more higher-resolution seismic imaging results (Yang et al., 2022).

(2) The difference in amount of pre-existing weak zones. Sun et al. (2022) suggested that the Indian push due to India-Asia collision reactivated the pre-existing weak zones inherited from previous closure of the South Tianshan Ocean and forced the western Tarim lithosphere subduct northward. Our simulation results also show that local weak zone can facilitate lithosphere subduction (Figure 10). Thus, if there are more pre-existing weak zones or faults in the central and western Tien Shan than the eastern Tien Shan, it may cause their different deformation mode.

Of course, these are all just our guess, and more observations and studies are needed.

**7.** Finally regarding the results figures: please do not use the rainbow (or jet) colormap! I know this might sound picky, but there is strong science behind this request. See Crameri et al. 2020: The misuse of colour in science communication. Crameri also provides a wide range of scientific colourmaps in a wide range of formats. See Crameri, 2018: http://doi.org/10.5281/zenodo.1243862.

Responses #7: We changed the color bar of viscosity to batlow which has better ability to distinguish details. Please see Figures 3-6, 10 and Figure S2.

**Other comments:**

**8.** Lines 21-22: Do they vary somehow systematically with the different rheological characteristics chosen for the models?

Responses #8: Yes, we listed all the simulation results in Figure S1 and S2, and summarized and discussed the relationships of rheological characteristics of the Pro-, Mid- and Retro-terranes and different deformation modes in Figure 8 and section 4.1.

**9.** Line 20: "…studied the effects of different rheological assumptions terrane deformation"— I understand what the authors are trying to say but there is something missing here in this sentence.

Responses #9: Thanks for your reminder. We added an "on" in this sentence as "…*studied the effects of different rheological assumptions on terrane deformation*". Please see Line 20.

**10.** Lines 21-23: I am not sure that this blanket statement adds anything new to the abstract.

Responses #10: We deleted this sentence. Please see Lines 21-23.

**11.** Line 32: Here first the authors talk about accreted terranes than about cratons than finally about continental fragments.

Responses #11: Continents are composed of many secondary terranes of different ages, such as ancient cratons and young continental fragments. These terranes have various lithospheric strengths due to differences of ages, components and structures. In this paragraph, we mainly wanted to express that the strength of the continental lithosphere is strongly laterally heterogeneous.

**12.** Line 77: I assume the implications of this study should be taken as global implications, and not specific to Eurasia even though the natural examples selected are from Asia. I would advise the authors to make that distinction clear in the introduction.

Responses #12: Our simulation results can be implied to the global regions, especially in East Aisa where there are two typical ongoing intracontinental orogens of the Tien Shan orogenic belt and Tibetan Plateau and an ancient orogeny of the Early Paleozoic Orogen in Southeastern China. We have stated this in the Introduction, please see Lines 78-80.

**13.** Line 78: The methodology is sorely missing a limitations sub-section. This is problematic, especially in the light of the supplementary models involving structural inheritance.

Responses #13: We added "Model Limitations" as section 5, in which we stressed the potential effects of pre-existing weak zones, lithosphere structure and convergence rate. Please see Lines 409-424.

**14.** Lines 131-132: This is unclear. What are these rates dependent on? I assume they are not uniform everywhere under every circumstances otherwise they would make no sense at all. Please clarify this in the description.

Responses #14: We added detailed descriptions about the sedimentation and erosion rates, please see Lines 135-137.

**15.** Line 135: "grids" or "grid" is "grid cells". The grid is the whole thing and it consists of grid-cells. They are non-uniform, but rectangular, correct?

Responses #15: Yes, they were corrected, please see Line 140. Thank you very much.

**16.** Lines 147-148: What was the basis of these choices? What do these thicknesses correspond to? The 160 km feels like a cratonic to orogenic lithosphere to me but than is it reasonable to assume that the same crustal structure would sit on top of it that is part of the thinner lithospheric structures?

Responses #16: In order to reflect the heterogeneous thickness of continental

lithosphere (Pasyanos et al., 2014), we set different thicknesses of the lithospheric mantle for the three terranes in our models. Their thickness was chosen arbitrarily. We discussed the effects of lithospheric thickness on deformation in section 4.2 and also emphasized this in the Model Limitations. At the same time, for the purpose of simplification, we set the three terranes to the same crustal structure. We have rewritten this part in section 2. Please see Lines 150-156.

**17.** Line 159: "set to $0\,°C$".
Responses #17: Corrected.

**18.** Lines 161-164: Are these initial thermal conditions stable over time, or would they change even in the absence of any deformation?
I have a feeling that all three of them cannot be all stable at the same time. This would lead to Moho-temperatures that evolve over time regardless of whether they undergo deformation or not.
Actually, the crustal heat-production means that they are definitely not stable.
This is not a deal-breaker, but it does make the interpretation of the model-results a little-bit harder.
Responses #18: The temperature filed will mildly evolve over time even if these terranes do not deform (Figure R1).

**19.** Lines 167-168: This is part of the mechanical boundary conditions and not the thermal ones so I would strongly suggest moving it up within this paragraph. I would also argue that the vertical boundaries are free-slip in the vertical direction but have fixed velocity boundary conditions in the horizontal direction. I would also like to see written down what happens in the vertical direction at the top boundary.
Responses #19: The description about the setup on convergence rate was move to the part of mechanical boundary conditions. Please see Lines 165-166.
The convergence rate was applied within the model, so it is internal boundary condition which would not affect the boundary conditions for free slip on both sides.
The model top is a free-slip boundary which means that velocity in the vertical direction at the top boundary is zero. We added this to the manuscript, please see Lines 160-161.
In addition, we set a "sticky air" layer between the model top boundary and top of the upper crust. The interface between "sticky air" layer and the upper crust can be regarded as free surface which allow the evolution of topography.

**20.** Lines 173-174: It would be very useful if a working definition for these modes would be defined here. What were the criteria used to categorize the behavior of the individual experiments? What emerging features of particular behavior decided whether an experiment is categorized as subduction or collision?
Responses #20: We added the main deformation features of different deformation modes to distinguish them before describe their detailed evolutional processes. Please see Lines 185-193.

**21.** Lines 185: "opposite dip directions", Opposite to what direction?

Responses #21: To avoid misunderstanding, we have deleted this expression. Please see Line 205.

**22.** Line 190: "flower-like"—Are you describing the state of the experiment at 24.2 Myr? Three snapshots are provided on figure 3 but they are never individually referenced in the text which I think is a missed opportunity. It would help guiding the reader through this passage. This comment applies to all the following model descriptions as well.

Also, if it is indeed the state of the experiment at 24.2 Myr, than maybe it would be worthy marking this structure on the figure itself? I think I have an idea what the authors mean, but clarity would be helpful.

Responses #22: We labeled each snapshot in Figures 3-6, and individually referenced them in the text. We also pointed out the "flower-like" structure in Figure 3(b) whose evolutional time is 24.2 Myr.

**23.** Line 192: "Ultimately"—This is the final snapshot, correct? Why is this not termed as subduction of the pro-terrane?

Responses #23: The Pro- and Retro-terrane's lithospheric mantles meet and collide together in this scenario. So, we define it as collision.

**24.** Lines 195-197: After circling back from the bottom of section 3, is this supposed to be a similar concluding description of the criteria used to identify experiments falling into this category as provided for the other 3 cases? In that case, I would suggest reformulating and clarifying it as I missed it as such on the first pass.

Responses #24: We added the main deformation features of the four deformation modes to distinguish them before describe their detailed evolutional processes. So, we deleted this paragraph here. Please see Lines 216-218.

**25.** Line 201: "weaker lithospheric mantle" —This weakness is due to the different LAB depth and through that, the different thermal structure, correct? Maybe it would be useful to spell this out here?

Responses #25: Yes! We added this description in the text, please see Lines 223-224.

**26.** Line 206: "inducing shortening and thickening" —I find it a bit difficult to tell from the snapshots: is the thickening in the upper crust achieved primarily by folding or faulting? Is upper-crustal deformation completely decoupled from the lower-crustal folding or are they connected? Perhaps some strain-rate contours could help answer that question.

I am just curious here. I am conscious that this has probably little bearing on the large-scale patterns discussed by the manuscript but maybe a few words thrown in

here would not increase clutter too much.

Responses #26: Shortening and thickening of the upper crust in the Mid-terrane can be clearly saw in Figure 4b and 4c. Deformation mainly occurs in the upper crust, while the strong lower crust of the Mid-terrane almost keeps undeformed (Figure R3). We added related description in the text, please see Line 230.

[Figure]

Figure R3. Crustal deformation and topography uplift of deformation mode of lithosphere subduction. The upper, middle and lower panels are topography, component and strain rate results, respectively.

**27.** Line 208: "and scrapes", I would suggest replacing these two words with the word "scraping".

Responses #27: Corrected. Please see Line 232.

**28.** Lines 214-216: This is largely the type of criteria-definition I was missing from around line 173.

Responses #28: We added the main deformation features of the four deformation modes to distinguish them before describe their detailed evolutional processes. So, we deleted this expression here. Please see Lines 239-241.

**29.** Line 220: "fragile", in my mind the word "fragile" is strongly associated with brittle deformation. I would actually use the phrase rheologically weaker here.

Responses #29: Corrected. Please see Line 245.

**30.** Line 222: "in which leads to crustal folding…", Does the ductile thickening of the mantle-lithosphere really lead to crustal folding? Looking at the viscosity plots I would argue that the top of the mantle-lithosphere deforms in a plastic manner, initially localizing in two discrete zones, out of which one will develop into a subduction zone by the end. These zones of deformation localize then the initial crustal folding but by 19.9 Myr the entire Pro-terrane crust seems to be folding.

Responses #30: To describe the simulation results more accurately, we rewrote this sentence as "*The lithosphere of the Pro-terrane is first thickened, and crustal folding is formed in two discrete zones*". Please see Lines 246-248.

**31.** Line 223: "After", I am not super-certain I would agree with that either. Delamination is not complete at 19.9 Myr but subduction initiation seems to have already been achieved.

Responses #31: Delamination is completed at ~26 Myr, before which the Pro-terrane absorbs plate convergence in the form of thickening, and then mainly subduction. We rewrote this sentence as "*After delamination of the thickened lithosphere, subduction of the Pro-terrane's lithospheric mantle along one of the deformation localization zones absorbs the plate convergence*". Please see Lines 250-252.

**32.** Line 225: I am curious to know if this is a feature of all the Thickening and Delamination models and whether the authors have an explanation for this particular feature.

Responses #32: There was a clerical error, and we have corrected as "*Crustal deformation is restricted in the Pro-terrane until lithosphere delamination…*" This feature appears in all the Thickening and Delamination models, and we explained it in section 4.1.

**33.** Line 231: So, why is CB-II, CB-I, CB-I (Retro-Mid-Pro; figure S1a; I will use this way of referring to experiments in all my comments) thickening and delamination and not subduction? I do not really see on that one snap-shot any sign of significant mantle-lithospheric dripping (or delamination). The same goes for CB-II, CB-I, JS-I and CB-II, CB-I, JS-II (same figure).

In fact, a lot of the time, I can't tell from the snapshots what were the reason for placing a particular experiment into the "thickening and delamination" case or the "collision" case and not into the subduction case.

Is it purely because the other bounding terrane remains largely undeformed in these experiments? I would argue that that fact alone does not make the observed behavior delamination as opposed to subduction.

Responses #33: The detached Pro-terrane's lithospheric mantles in models of CB-II/CB-I/CB-I, CB-II/CB-I/JS-I, and CB-II/CB-I/JS-II sank deeper into the mantle (Figure R4). They may be not shown in Figure S1a due to the smaller vertical range selected when drawing the picture. The Pro-terrane has an extremely weak lower crust

and lithospheric mantle, causing it to be easily thickened when it suffers from compression.

In order to better distinguish experimental results, we added the main deformation features of the four deformation modes in section 3 and deleted this part. Please see Lines 185-193 and 256-260.

[Figure]

Figure R4. Delamination of the Pro-terrane's lithospheric mantle. The left, middle and right panels were the component results of models of CB-II/CB-I/CB-I, CB-II/CB-I/JS-I, and CB-II/CB-I/JS-II, respectively.

**34.** Lines 235: "Pro-terrane", Based on the supplementary figures, this can also be the retro-terrane, so I would suggest using the phrase bounding terrane here.

Responses #34: We described the simulation results of Case 4 in section 3.4. Therefore, we believe that "Pro-terrane" can more accurately describe the specific results shown in Figure 6.

**35.** Lines 241-243: Again, I would interpret this experiment differently. To me, it looks like the very weak bottom portion of the Pro-terrane mantle-lithosphere drips. It seems to be an order of a magnitude weaker then the underlying sub-lithospheric mantle, so this is not a huge surprise. In the mean time, the Mid- and Retro-terrane mantle-lithosphere under-thrusts the strong upper portion of the Pro-terrane mantle-lithosphere, initiating continental subduction.

Responses #35: Yes, the Pro-terrane's lithospheric mantle is so weak that it can be easily scraped off by the strong Mid-terrane's lithospheric mantle. After that, the strong Mid-terrane's lithospheric mantle underlies sub-lithospheric mantle and replaces the primordial lithospheric mantle of the Pro-terrane. So, we termed this deformation mode as replacement. Please see Lines 191-193.

**36.** Lines 245-248: Again, it is nice that the authors provide a short guide to what behaviors were used to classify the individual models, but I would rather have these together for all 4 cases. Note, that such description is not provided for the collision case.

I would also like to see an explanation as to why were these particular criteria used and an argument as to why they are meaningful.

Responses #36: We added the main deformation features of the four deformation modes to distinguish them before describe their detailed evolutional processes. So, we deleted this expression here. Please see Lines 274-277.

**37.** Line 251: "results" is "result", May I suggest the word arise rather than result? Result is not wrong, just a bit confusing in my opinion.
Responses #37: Corrected. Please see Line 280.

**38.** Line 252: "Figure 8", I would like to commend the authors on providing Figure 8 as well as Figure S1. It is very nice to have the parameter-space figure backed up by a supplementary figure showing actual model-snapshots.

The only way to significantly improve on this would be to provide an additional supplementary figure, analogue to S1, but showing the viscosity field of the experiments.
Responses #38: Thank you very much. The viscosity fields of all experiments were provided in Figure S2 in the Supplements file.

**39.** Line 255: "lithospheric mantles of its surrounding Pro- and Retro-terranes", How come this behavior is there when the Mid-terrane mantle-lithosphere is strong but its lower-crust is weak (JS-II)?

The authors' explanation explains a large portion the experiments classified as "Collision" - 19/24 if I count the ones where the mid-terrane rheology is CB-I, but there are still 5 models where the mantle-lithosphere of the mid-terrane is definitely not weaker than the mantle-lithosphere of the surrounding terranes. As a side-note, the experiment presented as a type-example of this behavior is one of the 5 models where this explanation is not valid.
Responses #39: Please refer to Responses #4.

**40.** Lines 255-257: I would suggest re-phrasing this sentence; I understand the meaning of it, but it is grammatically awkward.
Responses #40: This sentence was rewritten as "*When one of the two bounding terranes has extremely weak lithospheric mantle, its lithosphere is first to be thickened by compression, and delamination may follow due to density-driven instability*". Please see Lines 285-287.

**41.** Line 257: I would add here how this is also apparent on figure 8 with 3 of the four panels showing an upper-left cross-like pattern (grey circles when the Retro- or the Pro-terrane has a rheology of CB-II). This could also be somehow highlighted on figure 8.
Responses #41: We connected these gray circles by two cross-like solid gray lines to highlight them. Please see Figure 8.

**42.** Line 260: Again, I would point out how nicely this shows up on figure 8 (red squares along the lower and right edges of the two left panels).

Responses #42: Thank you very much. We connected these red squares by two solid red lines to highlight them, too. Please see Figure 8

**43.** Lines 261-262: I assume the Authors' mean strong lower crust and Weak lithospheric mantle?

Responses #43: Yes, thank you very much. We have added the "weak", please see Line 292.

**44.** Line 263: Again, point to figure 8 (stars along the top and left edge of the bottom right panel).The pattern is appealing, but as discussed above, I do have reservations about the description of the observed behavior.

Responses #44: Thank you very much. We were also very surprised when we saw this parameter-space figure after summarizing all the simulation results according to their main deformation characteristics. This also indicates that our simulation results can indeed reflect the features of lithosphere deformation in a collisional system with multiple terranes to a certain extent.

**45.** Liens 265-266: This seems mostly true, although there are cases where this is a stretch. See for an example CB-I, JS-I, CB-I on figure s1c, CB-I, JS-II, CB-II on figure s1d or the oddly thickened Pro-terrane mantle-lithosphere of JS-II, JS-I, JS-I on figure s1c.

This latter example in particular seems peculiar to me as there appears to be no thickening of the Pro-terrane crust, so where does this thickening of the mantle-lithosphere comes from?

In fact, the Pro-terrane mantle-lithosphere seems to have a tendency to thicken even when there is not much crustal shortening above it. CB-II, JS-II, CB-II and CB-II, JS-II, JS-II seem to have similar amounts of Mid-terrane crust thrust on top of the Pro-terrane mantle-lithosphere but appear to have significantly different mantle-lithosphere thickening.

Am I wrong? Could the Authors explain these observations?

Responses #45: The deformation mode of the model of JS-II/JS-I/JS-I is subduction rather than thickening and delamination. We made a wrong label in Figure S1c, and have corrected it. Please see Figure S1c.

In addition, in the models of CB-II/JS-II/CB-II and CB-II/JS-II/JS-II, the three terranes have same weak crustal structure but the Mid-terrane has strong lithospheric mantle. Thus, when the Pro-terrane (or Retro-terrane) was compressed, the strong lithospheric mantle of the Mid-terrane can block the advance of weak lithospheric mantle of the Pro-terrane (Retro-terrane), but crustal shortening can propagated into the Mid-terrane. This means that crustal convergence was absorbed by the Pro- (or

Retro-) and Mid-terranes, but lithospheric mantle convergence was only absorbed by the Pro-terrane (or Retro-terrane), which can explain the lithospheric mantle of the Pro-terrane (or Retro-terrane) seems to have a tendency to thicken even when there is not much crustal shortening above it.

**46.** Lines 272-273: I had to read this sentence several times and I am still unsure what it means. It feels like a very vague synopsis of the manuscript. Please either clarify or remove this sentence.
Responses #46: We deleted this sentence. Please see Lines 302-303.

**47.** Line 280: "Figure 9", It would really help the reader if the reference examples would also appear on this figure.
Responses #47: Final simulation results of the four reference examples were added as Figure 9a.

**48.** Line 281: "three terranes is comparable"—Would it be possible the get the theoretical strength-envelopes of the individual terranes all plotted on the initial-setup panels of figure 9?
Responses #48: Theoretical strength-envelopes of the individual terranes in Cases 1and 2 were shown in Figures 3g and 4g, respectively. Now, Figure 9 contains 25 subplots and is very complicated. It will become much more tanglesome if plotting all strength-envelopes of the individual terranes on the initial-setup panels in Figure 9. Thus, we did not add the strength-envelopes on it.

**49.** Line 283: Please expand on this. What is distinct about the new patterns? I still find it hard to distinguish between the collisional case and the subduction case based on a single time-frame of an experiment.
Responses #49: We compared in detail the main differences in simulation results of Cases 1 and 2 arising from changes in lithospheric thickness. Please see Lines 327-332.

**50.** Lines 285-286: Delate "with ta weak mantle Crème brûlée rheology".
Responses #50: Corrected. Please see 335.

**51.** Line 291: "In addition"—Had this exercise been done for all experiments? Why is the weak-zone assumed to tilt towards the mid-terrane rather than away from it?
Responses #51: Since the local weak zone is not the focus of this study, we only tested its effects on lithospheric deformation for Cases1-4, and the dipping direction of the weak zone is also arbitrary. We believe that these comparative experiments are sufficient to illustrate the important influence of the weak zone in the deformation of the lithosphere. As for how it changes the deformation of the lithosphere, this may need to be further explored in new studies.

**52.** Lines 296-298: This is a useful side-note to contextualize the presented results.

Based on the limited results presented here, the nature of the contact between the terranes has a first order effect on their deformation, regardless of their rheological structure which is not surprising.

In any case, these additional model results display an important constrain on how much we can read in to the rest of the experiments. It is useful to show that we can produce a range of observations purely by varying the rheological model of the different terranes in play but these effects might be partially or fully overridden by inherited structural constraints.

I do think that this assessment should be part of the discussion.
Responses #52: We added the simulation results about the effects of the local weak zone on lithosphere deformation to the manuscript as Figure 10.

**53.** Line 299: Why isn't there a section provided for each of the natural systems used for comparison? It is a bit strange that the eastern Tien Shan gets several panels, but neither the Tibetan Plateau nor the Early Paleozoic Orogen of Southeastern China gets any visual representation.
Responses #53: We divided section 4.3 into three small parts, and applied the simulation results to the eastern Tien Shan, the Tibetan Plateau and the Early Paleozoic Orogen in Southeastern China, respectively. Please see 350-408.
At the same time, we added the cross-sections of the Tibetan Plateau and the Early Paleozoic Orogen in Southeastern China in Figure 11.

**54.** Line 315: Based on the seismic tomography study used and cited here, there is considerable variation in the deep behavior of the Tarim and Junggar basins and the Tien Shan along an east-west transect (from subduction to "upwelling" to collision). In the Author's opinion, could this be attributed to changes in rheology or something else?
Responses #54: Please refer to Responses #6。

**55.** Line 333: "Sn", This should be a sub-script. Also, please explain a bit what inefficient Sn propagation. In my opinion, it is a fairly specific seismological term so not everyone might be familiar with it.
Responses #55: Sn is a seismic shear wave which propagates in the high-velocity mantle below the crust and above low-velocity zones. Efficient propagation is characteristic of Sn, for much of the Earth's surface, especially shields, stable continental patforms, and ocean basins. The inefficient propagation of Sn is thought to be due to attenuation in the mantle, associated with a thin or absent lithospheric mantle lid between the crust and asthenosphere. We explained it in the text, please see Line 386.

**56.** Line 595: At the moment, the individual locations cannot be easily identified on the map. I realize, that this might be difficult to achieve in an aesthetic way, but

maybe numbering the dots on the map and the strength profiles could be a solution?

Responses #56: We numbered the dots on the topography map and labeled them on corresponding strength profiles, which really help to identify the individual locations of strength profiles. Please see Figure 1.

**57.** Line 622: Please do not use the rainbow (or jet) colormap! This suggestion stands for all the figures. See Crameri et al. 2020: The misuse of colour in science communication. Crameri also provides a wide range of scientific colourmaps in a wide range of formats. See Crameri, 2018: http://doi.org/10.5281/zenodo.1243862. Please add a) b) c) etc. to the panels. This suggestion stands for all the results figures.

Responses #57: We changed the color bar of viscosity results to batlow, and labeled each snapshot with (a), (b), (c) …for all results figures. Please see Figures 3-6.

**Comments about the Supplements:**
**58.** Figure S1—Please do number these panels so that one can easily reference them in writing.

Responses #58: We have numbered each of panels in Figure S1. Please see Figure S1.

**59.** Figure S1c—This is the type-example given in the core text of subduction, yet it is marked as thickening and delamination here.

Responses #59: This incorrect marking was a clerical error and we have corrected it. Please see Figure S1c.
* * *
**Comments from Referee #2:**
Review on "Various lithospheric deformation patterns derived from rheological contrasts between continental terranes: Insights from 2-D numerical simulations" by Xie et al.

It's good to do the systematical study testing the effects of various rheological structures on continental collision. The modeling results are robust. My suggestion is minor revision.

**Comments:**
**60.** Vertical rheological structures are highlighted in figure 1. But it seems that the horizontal rheological variation is a main feature of the model setup (e.g., figure 2). One should discuss which one plays the dominant role in collisional dynamics?

Responses #60: In this study, the horizontal rheological variation of different terranes in a collisional system is our research emphasis. From our simulation results, it is difficult to determine whether the horizontal strength contrasts between terranes or the vertical strength variation of a single terrane plays the dominant role in a multi-terrane collisional system. This is also the significance and necessity of our study. We added some related discussions in section 4.1. Please see Lines 304-316.

**61.** Figure 1: I like the plot that showing the variations in the vertical rheological

structures based on natural examples. However, in my opinion, one should describe how these rheological structures are plotted and why they are different.

Responses #61: Thank you very much. We briefly described the method of rheological calculation on the caption of Figure 1, and pointed out that different temperature and lithospheric compositions at different sites lead to a diverse suite of strength profiles vs. depth. Please see Lines 677-682. More details of the rheological calculation can be found in the references.

**62.** Figure 2: Horizontal rheological variation due to the presence of the weak terrane is a big feature in the model setup. In my opinion, the setup does not link to the motivation (figure 1) tightly. It will be good to explain why employ the multiple-terrane setup as well as the particular thermal structures.

Responses #62: We added "*Large-scale continental collisional system often involves the multiple units of an indenting terrane, a middle terrane, and far-end backwall terranes. These terranes have different lithosphere rheologies and thicknesses, and they collectively contribute to several styles of continental deformation (Artemieva, 2006; Audet and Bürgmann, 2011; Pasyanos et al., 2014; Morgan and Vannucchi, 2022)*" in the introduction. Please see Lines 68-72.

Such a statement, together with the explanation in the model setup (please see Lines 148-155) and a discussion of effects of lithospheric thickness (section 4.2), makes the article logically coherent and its parts tightly linked.

**63.** Figure 3: The pro-terrane subducted, because it's colder and thicker? Is it always like this?

Responses #63: Figure 3 shows the simulation results of model of JS-I/JS-II/JS-I. In models CB-I/JS-II/JS-I, CB-II/JS-II/JS-I, the lithosphere of the Pro-terrane is also colder and thicker which is same with that in model of JS-I/JS-II/JS-I, but they respectively show deformation modes of lithosphere replacement and lithosphere thickening and delamination owing to their weaker lithospheric mantle of the Pro-terrane (subplots 3, 7 in Figure S1d). Therefore, we suggest that the strength of the lithospheric mantle plays a more important role in lithosphere underthrusting of the Pro-terrane.

**64.** Line 626: Space is missed between the number and the unit.
Responses #64: Corrected.

**65.** What are the red arrows in the result plots (figures 3-6)? Convergence?
Responses #65: The arrows in figures 3-6 indicate the direction of convergence. To avoid misleading, we set all arrows to the same length. Please see Figure 3-6.

**66.** Figure 8: I guess the velocity boundary condition may affect the model results. How will model results change if the pushing velocity is imposed on the right side? Or both sides?
Responses #66: If the convergence rate is placed on the right or both sides of the

model, although there will be subtle differences in the simulation results, the final deformation mode is the same. Please see Figure R5.

[Figure]

Figure R5. Effects of the direction of the convergence rate. (a), (b) and (c) are the final simulation results of models with different convergence directions, and (a) is the final simulation results of Cases 1-4 (reference examples).

**67.** Figure 10: Is the model designed particular for this region? If yes, one should mention in the introduction. Otherwise, I may ask why the particular model setup is used (e.g., thicker and colder pro-terrane).

Responses #67: Our models can be implied to the global regions, especially in East Asia where exist two typical ongoing intracontinental orogens of the Tien Shan orogenic belt and Tibetan Plateau and an ancient orogeny of the Early Paleozoic Orogen in Southeastern China. We stated this in the Introduction, please see Lines 77-80. We also added the profiles of Tibetan Platea and the Early Paleozoic Orogen in Southeastern China in Figure 11.

Lithosphere thickness of the Pro-, Mid- and Retro-terranes was chosen arbitrarily. We discussed the effects of lithospheric thickness on deformation in section 4.2 and also emphasized this in the Model Limitations.

---

## Referee Report (RR1)

The manuscript titled „Various lithospheric de formation patterns derived from rheological contrasts between continental terranes: Insights from 2-D numerical simulations" by Renxian et al. addresses the topic of continental collision-dynamics in the presence of multiple terranes using two-dimensional thermo-mechanical numerical experiments.

This is my second round on the manuscript and I am generally happy to accept the changes to the text and the arguments put forward by the Authors. I only have minor comments about the modified text regarding the language that I picked up on and one more pertinent issue with the comparison to natural systems. I will start with this latter:

Adding extra cross-sections and improving the map helped quite a lot with this aspect of the text, but the comparison to South-eastern China still only references a solitary paper from 2009 and gives very little context. I am ever so sorry to write this, but I would strongly suggest the Authors to revisit this section and at least slightly expand on it.

Minor comments on the language:

Line 76-78: "apply the simulations to better understand on going and past deformation histories of various orogenic belts in the global, especially in eastern Asia": in the global makes no sense in this sentence. I would strongly suggest rephrasing this.

Lines 133-135: So there is sediment in the model-domain everywhere below 5 km? With the low sedimentation-rate employed, this is probably not a big deal, but I would like a line on justifying this choice of parametrization.

Lines 162-163: Nothing to be done here, I just wanted to note to the Authors, that Now I understand the boundary condition employed. I feel a bit silly that this was unclear for me the first time around.

Line 234: "starts to form folding" I would remove the word form.

Line 243: I would reintroduce the removed "how" into this sentence to make it grammatically correct again.

Line 387: I think influence should be singular here.

Line 391: "chosen" instead of "chose" and "they also have" instead of "they also has" would be the grammatically correct wording.

Lines 392-396: I think "some studies believe" is an inappropriate phrasing. These studies made scientific arguments. Belief has not much to do with that. Furthermore, the authors did not explore different convergent velocities (as far as I know) so stating that "the impact of the convergence rate

almost can be ignored" is just plain wrong. Unless the Authors have tested the model-behaviours for different velocities, they should just acknowledge that varying these parameters was beyond the scope of the study and have not been carried out. There is nothing wrong with that.

Figures: I am happy you have changed the colormaps. I would suggest acknowledging the source of the colormaps in the Acknowledgements section.

---

## Author Response (AR2)

**Comments of** "*Various lithospheric de formation patterns derived from rheological contrasts between continental terranes: Insights from 2-D numerical simulations*"
* * *
**Comments from Referee #1:**

**Suggestions for revision or reasons for rejection**
(visible to the public if the article is accepted and published)
The manuscript titled "*Various lithospheric de formation patterns derived from rheological contrasts between continental terranes: Insights from 2-D numerical simulations*" by Renxian et al. addresses the topic of continental collision-dynamics in the presence of multiple terranes using two-dimensional thermo-mechanical numerical experiments.

This is my second round on the manuscript and I am generally happy to accept the changes to the text and the arguments put forward by the Authors. I only have minor comments about the modified text regarding the language that I picked up on and one more pertinent issue with the comparison to natural systems. I will start with this latter:

**(1)** Adding extra cross-sections and improving the map helped quite a lot with this aspect of the text, but the comparison to South-eastern China still only references a solitary paper from 2009 and gives very little context. I am ever so sorry to write this, but I would strongly suggest the Authors to revisit this section and at least slightly expand on it.
Responses #1: Thanks for your suggestions. We added more comparisons and references about the Early Paleozoic Orogen in Southeastern China to enrich our discussion in section 4.3.3. Please see Lines 372-394.

Minor comments on the language:

**(2)** Line 76-78: "apply the simulations to better understand on going and past deformation histories of various orogenic belts in the global, especially in eastern Asia": in the global makes no sense in this sentence. I would strongly suggest rephrasing this.
Responses #2: We deleted "in the global", please see Line 77.

**(3)** Lines 133-135: So there is sediment in the model-domain everywhere below 5 km? With the low sedimentation-rate employed, this is probably not a big deal, but I would like a line on justifying this choice of parametrization.
Responses #3: In our models, topography is calculated by subtracting the initial position (e.g., Y=20 km in our models) from the current vertical position of the surface. If the topography is convex upward and its height is higher than 5 km, then it will suffer erosion with a rate of 0.3 mm/yr; otherwise, if the topography is depressed downward and its depth is lower than 5 km, then it will undergo sedimentation with a

rate of 0.3 mm/yr.

Owing to surface processes are not our focuses in this study, for the aim of simplification, we set a small erosion and sedimentation rates of 0.3 mm/yr, which are similar to previous studies (Gerya and Yuan, 2003b; Bian et al., 2020). As well, we simply choose a very large value of 5 km as the threshold for initiating denudation and sedimentation to further weaken the influences of surface processes on the evolutions of our model. We added some explanations about the choices of these parameters. Please see Lines 136-140.

**(4)** Lines 162-163: Nothing to be done here, I just wanted to note to the Authors, that Now I understand the boundary condition employed. I feel a bit silly that this was unclear for me the first time around.
Responses #4: Thank you very much.

**(5)** Line 234: "starts to form folding" I would remove the word form.
Responses #5: It was corrected as "starts to fold". Please see Line 240.

**(6)** Line 243: I would reintroduce the removed "how" into this sentence to make it grammatically correct again.
Responses #6: Thank you very much. We re-add "how" in this sentence, please see Line 249.

**(7)** Line 387: I think influence should be singular here.
Responses #7: Yes, sometimes those local preexisting weak zones may control the lithospheric deformation.

**(8)** Line 391: "chosen" instead of "chose" and "they also have" instead of "they also has" would be the grammatically correct wording.
Responses #8: They were corrected. Please see Line 411, thank you very much.

**(9)** Lines 392-396: I think "some studies believe" is an inappropriate phrasing. These studies made scientific arguments. Belief has not much to do with that. Furthermore, the authors did not explore different convergent velocities (as far as I know) so stating that "the impact of the convergence rate almost can be ignored" is just plain wrong. Unless the Authors have tested the model-behaviours for different velocities, they should just acknowledge that varying these parameters was beyond the scope of the study and have not been carried out. There is nothing wrong with that.
Responses #9: We changed "believe" to "suggest" so that it can express what we mean more accurately. Please see Line 412.
In addition, we also pointed out that we did not discuss the impact of convergence rate in this study in section Model Limitations, please see Line 417.

**(10)** Figures: I am happy you have changed the colormaps. I would suggest acknowledging the source of the colormaps in the Acknowledgements section.

Responses #10: Thank you very much. We acknowledged Crameri et al. (2018) for using their color bar of batlow in the figures of viscosity field. Please see Lines 454-455.
* * *
**Comments from Referee #2:**

**Suggestions for revision or reasons for rejection**
(visible to the public if the article is accepted and published)

Responses: We are grateful to reviewer #2's the insightful comments on improving our manuscript, and we are also grateful to reviewer #2 for the recognition of our first round of revisions.